# Inversion of accommodation zones in salt-bearing extensional systems: insights from analogue modeling

Elizabeth P. Wilson, Pablo Granado, Pablo Santolaria, Oriol Ferrer, Josep Anton Muñoz

Institut de Recera Geomodels, Departament de Dinàmica de la Terra i de l'Oceà, Universitat de Barcelona, 08028, Barcelona, Spain

*Correspondence to*: Elizabeth P. Wilson (epwilson@ub.edu)

**Abstract.** This work uses sandbox analogue models to analyze the formation and subsequent inversion of a decoupled extensional system comprised of two segmented half-grabens separated by a diffuse accommodation zone with thick early syn-rift salt. The segmented half grabens strike perpendicular to the direction of extension and subsequent shortening. Rifting created first a basement topography that was infilled by model salt, followed by a second phase of extension and sedimentation, followed afterwards by inversion. During the second phase of extension, syn-rift syncline minibasins developed above the basement extensional system and extended beyond the confines of the fault blocks. Sedimentary downbuilding and extension initiated the migration of model salt to the basement highs, forming salt anticlines, reactive diapirs, and salt walls perpendicular to the direction of extension, except for along the intervening accommodation zone where a slightly oblique salt anticline developed. Inversion resulted in decoupled cover and basement thrust systems. Thrusts in the cover system nucleated along squeezed salt structures and along primary welds. New primary welds developed where the cover sequence touched down on basement thrust tips due to uplift, salt extrusion, and syn-contractional downbuilding caused by loading of syn-contractional sedimentation. Model geometries reveal the control imposed by the basement configuration and distribution of salt in the development of a thrust front from the inversion of a salt-bearing extensional system. In 3D, the interaction of salt migrating from adjacent syn-rift basins can modify the expected salt structure geometry, which may in turn influence the location and style of thrust in the cover sequence upon inversion. Results are compared to the northern Lusitanian Basin, offshore Portugal and the Isàbena area of the South-Central Pyrenees, Spain.

## 1 Introduction

Salt tectonics plays a key role in the development of extensional basins, such as the Permian and Triassic basins in Northwestern Europe (e.g., Ziegler and van Hoorn, 1989; Stewart & Coward, 1995; Stewart, 2007; Soto et al., 2017; Ahlrichs et al., 2020). Salt tectonics is also responsible for the characteristic structural styles of the circum-Mediterranean Alpine fold-and-thrust belts, such as the Ionian-Albanides (e.g. Velaj et al., 1999), the Tunisian Atlas (e.g. Saïd et al., 2011), the Swiss Helvetics (e.g. Sala et al. 2014) and Jura fold-and-thrust belt (Sommaruga, 1999), the southern Pyrenees (e.g. Muñoz et al., 2018), the Northern Calcareous Alps of Austria (e.g. Granado et al., 2019), or the SW France Sub-Alpine system (e.g. Célini et al., 2020). In salt-bearing extensional systems, the interaction between extensional basement faults and overlying salt layers acts as the primary control on the structural evolution of the post-salt cover system (e.g., Vendeville and Jackson, 1992; Vendeville et al., 1995; Withjack et al., 1990; Koyi et al., 1993; Withjack and Callaway, 2000; Dooley et al., 2003, 2005; Richardson et al., 2005; Fig. 1). During thick-skinned extension, salt migrates towards the fault plane in the downthrown block below the cover sequence, forcing the syn-rift depocenters to develop away from the fault plane, while extensional forced folds develop above the footwall block (Vendeville and Jackson, 1995; Withjack and Callaway, 2000; Richardson et al. 2005; Ferrer et al., 2014, 2016 and 2022; Tavani and Granado, 2015). Since rift basins are typically segmented by interacting basement faults and slip transfer systems, (e.g., Gibbs, 1984; Morley et al., 1990; Gawthorpe and Hurst, 1993; Coward, 1995) salt flow and associated sediment distribution may be significantly modified, departing from simpler 2D models. Evacuation of salt in

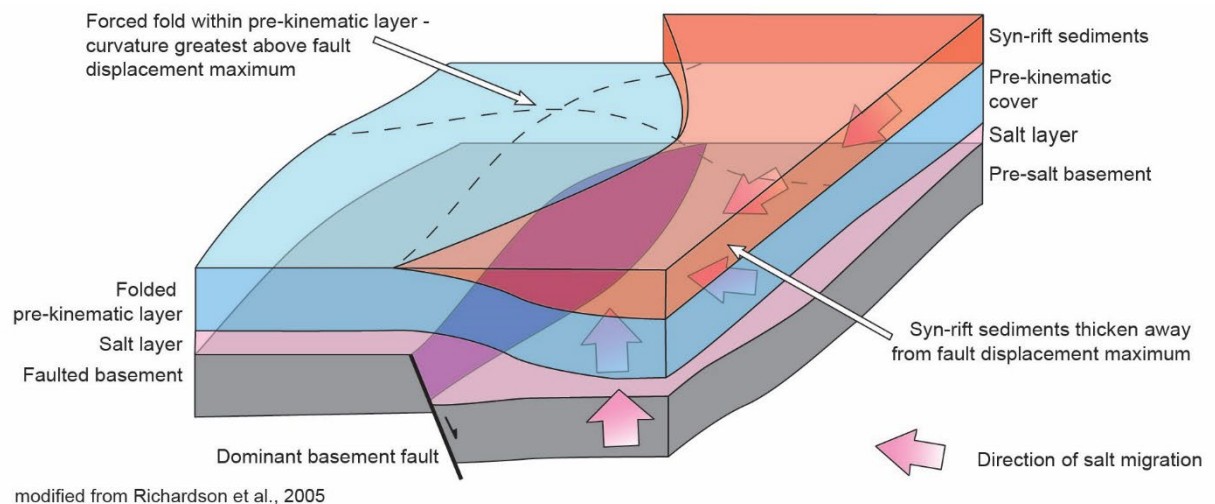

Forced fold within pre-kinematic layer - curvature greatest above fault displacement maximum

Syn-rift sediments
Pre-kinematic cover
Salt layer
Pre-salt basement

Folded pre-kinematic layer
Salt layer
Faulted basement

Syn-rift sediments thicken away from fault displacement maximum

Dominant basement fault

Direction of salt migration

modified from Richardson et al., 2005

**Figure 1. Block diagram illustrating the structural style of decoupled extensional rift half-graben with thick early syn-rift salt, modified from Richardson et al. (2015).**

rift basins by differential sedimentary loading upon crustal extension may significantly assist in creating accommodation space (e.g., Stewart et al., 1996, 1997; Dooley et al., 2003, 2005; Richardson et al. 2005), but also obscures the subsidence patterns of the pre-salt basement (e.g., Moragas et al., 2018; Strauss et al., 2021) as well as thermal records (Lescoutre et al., 2019).

When extension is coeval to salt evacuation produced by differential sedimentary loading, the geometry of the hanging wall syn-rift depocenters may strongly depart from classical wedge-like basins thickening into active basement faults, obscuring the pre-salt basement structural style (e.g., Richardson et al., 2005; Ferrer et al., 2016, 2022, Roma et al., 2018; Granado et al., 2021; Fig. 1). During shortening related to plate convergence, basement faults may be reactivated to different degrees, while the post-salt stratigraphy is usually sheared off from its autochthonous rifted basement. When this is the case, the original

relationships between the post-salt stratigraphy and the basement structures responsible for crustal thinning may not be directly observed (e.g., Callot et al., 2012; Mencós et al., 2015; Muñoz et al., 2018; Granado et al., 2019 and 2021; Snidero et al., 2019; Célini et al., 2020). Just as extensional faults play an important role during subsequent shortening, accommodation zones also may exert a strong control, leading to structurally complex 3D patterns of deformation, erosion, and sediment dispersal, as well as the associated salt flow patterns. To date, models of decoupled extensional rift basins with early salt do not illustrate

how salt below adjacent segmented half-grabens interacts across the intervening accommodation zones (Fig. 1).However, there are few works of natural case studies addressing how accommodation zones within segmented rift systems have been incorporated into contractional fold-and-thrust belts (e.g., Beauchamp, 2004; Mencós et al., 2015).This is of particular importance when these extensional accommodation zones are characterized by early salt structures, such as salt diapirs or salt sheets, that may alter the expected geometries and thickness and distribution of sediment across the accommodation zone.

There is a substantial number of scaled physical analogue modeling studies dealing with extension and inversion of rift basins (Koopman et al., 1987; Brun and Nalpas, 1996; Buchanan & McClay, 1991, 1992; McClay, 1989; 1995; 1996; Gartrell et al., 2005; Panien et al., 2005; Schreurs et al., 2006; Konstantinovskaya et al., 2007; Ferrer et al., 2016; Granado et al., 2017; Zwaan et al., 2022). Some programs modeled grabens and half-grabens over rigid basement blocks, which significantly force the basin inversion style (e.g., Dooley et al., 2005; Burliga et al., 2012; Ferrer et al., 2016, 2022; Moragas et al., 2017; Roma et al.,

2018a). Other experimental programs provide key insights into the styles of salt diapirs that develop along the margins of syn-extensional depocenters, the role of extensional salt structures in absorbing deformation early in shortening and the reactivation of welds as thrusts (i.e., *thrust welds*) during shortening of individual half-grabens (i.e., Dooley et al.,2003, 2005; Moragas et al., 2017; Stewart, 2017; Roma et al., 2018a; Hansen et al., 2021; Ferrer et al., *2022*; Miró et al., *2022*).

In this work we aim to investigate the tectonic inversion of segmented rift systems involving salt by developing an analogue modeling program. We have focused on the accommodation zones between stepped extensional faults, inspired by the Late Jurassic-Cretaceous rifted margin involved in the South-Central Pyrenees (Martín-Chivelet et al., 2019, a and b). It is important to note that although our model set-up does not show a change in fault polarity across the accommodation zone (*sensu* Morley ,1990), we have chosen to use this term because it also defines a style of diffuse fault displacement and not transfer along a single discrete element. The Pyrenean rifted margin has resulted from the superposition of rift systems (multi-stage and polyphase, Miró et al., 2021). An upper Triassic evaporitic unit was deposited at the end of a first Triassic rift event and was subsequently deformed by the main polyphase Late Jurassic-Early Cretaceous rift phase. During this main rift event the crust was hyperextended and consequently the mantle was exhumed and incorporated into the extensional system (Manatschal et al., 2021). The extensional systems was subsequently inverted during the Late Cretaceous-Paleogene contractional deformation as the Iberian and European plates collided (Muñoz, 1992; Manatschal et al., 2021). Many rifts and rifted margins along North Africa and Western Europe have followed a similar evolution, although with slightly different ages and rift maturity, and subsequently inverted during the Alpine orogeny. Examples are the Atlas Mountains, the Iberian margins, the SW Alps, and the Southern North Sea. These margins were superimposed on the Variscan orogenic belt and some of them involved external parts of the orogen characterized by deformed and slightly metamorphosed sedimentary successions. The analogue modeling program reproduces such multi-stage rifting before the tectonic inversion allowing the basement to be deformed. We have focused on the interaction between individual laterally segmented syn-rift basins across accommodation zones. In order to reproduce the desire geometry of stepped extensional faults involving the basement we have used silicone seeds embedded into a deformable coulomb analogue material, sand, similarly to previous studies (Le Calvez & Vendeville, 2002; Zwaan and Schreurs, 2017; Dooley and Hudec, 2020) We have followed a similar approach to Dooley and Hudec (2020), but with some significant differences that we will discuss below.

In this work, we modify the previous set-up of Dooley and Hudec (2020) to focus on the interaction of the salt and supra-salt sediment system across the accommodation zone between two half-grabens with an experimental program consisting of three analogue models in which extension is followed by contractional deformation, without and with syn-contractional sedimentation (see Tables 1 and 2). Our models aim to answer a series of questions related to the formation and deformation of laterally segmented salt-bearing rift basins: i) how does a late syn-rift salt layer decouple the supra- and sub-salt during subsequent extensional and contractional deformation events, ii) how does the salt control the depocenter distribution and geometry, iii) how does salt influence sediment distribution across the accommodation zone, iv) how does the inherited architecture of the sup- and supra-salt systems across the accommodation zone influence inversion and contractional deformation? Our modeling results are compared to natural systems characterized by different degrees of structural reactivation, and salt-detached shortening: the Northern Lusitanian Basin offshore Portugal or the Southern Pyrenees. Traditionally, these salt tectonic provinces have been important to the exploration and production of hydrocarbons. While this is still the case, these provinces are now of importance to the energy transition within the European Union, and across the globe (i.e., European Commission, 2021; Duffy et al., 2021; Duffy et al., 2023). Extensional salt tectonic provinces may provide the correct setting for the energy storage of hydrogen gas ($H_2$), compressed air, natural gas ($CH_4$), and the long-term storage of $CO_2$, while their inverted counterparts may provide untapped potential as sources of geothermal energy (e.g., Gasanzade et al., 2021; Martin-Roberts et al., 2021; Alves et al., 2022; Miocic et al., 2022).

**Table 1. Experimental models described in this paper.**

| Experimental Model | Modelled process | Total Extension | Total Shortening |
|---|---|---|---|
| Model 1 | Extension | 15 cm | |
| Model 2 | Extension + Shortening | 15 cm | 25 cm |
| Model 3 | Extension + Shortening with syn-contractional sedimentation | 15 cm | 25 cm |

## 2 Analogue Modeling Program

### 2.1 Experimental modeling materials and scaling parameters

The goal of the experimental program is to model the formation and contractional deformation of two salt-bearing half-graben basins segmented by an intervening accommodation zone. The models were constructed using modeling materials suitable to simulate upper crustal deformation (see Davy and Cobbold, 1991; Weijermars and Schmeling, 1986; Schellart, 2000; Dell'Ertole and Schellart, 2013), and following rigorous principles of dynamic scaling (Table 1). Dry well-sorted quartz sand (i.e., 98% pure silica) with an average grain size of 199 μm, a mean coefficient of friction (μ) of 0.6, and an average angle of internal friction (φ) of 34º, a bulk density of 1500 kg/m$^3$ and cohesive strength of ~55 Pascals was used (Ferrer et al., 2017). Sand displays an elastic/frictional plastic behavior with transient strain hardening prior to transition to stable sliding (e.g., Lohrmann et al., 2003; Adam et al. 2005), making it a reasonably good mechanical analogue of upper crust brittle rocks. The material used to simulate rock salt was Rhodosil GUM FB from Bluestar Silicones, a transparent viscous polydimethylsiloxane silicone polymer with a density of 972 kg/m$^3$ at room temperature and a viscosity of $1.6 \times 10^4$ Pa·s when deformed at an experimental strain rate of $1.83 \times 10^{-4}$ cm/s (Ferrer et al., 2016; Table 2). The polymer behaves as a near-Newtonian fluid having very low yield strength and a stress exponent $n$ of nearly 1 at the above-mentioned experimental strain rate. Near-Newtonian silicone polymer can be assumed as a reasonable first order approximation of salt rheology for analogue modeling experiments (Dell'Ertole and Schellart, 2013). Hereinafter, the term model salt will be used as a synonym to silicone polymer and the standard salt tectonics terminology used when describing polymer-related structures.

**Table 2.** Scaling parameters used in the experimental program.

| Quantity | Equation | Model | Nature | Scaling ratio |
|---|---|---|---|---|
| Thickness | | | | |
|     Sand overburden | | 20–60 mm | 2–4 km | $1.0 \times 10^{-5}$–$1.5 \times 10^{-4}$ |
|     Salt/polymer | | 10 mm (minimum) | 1000 m | $1.0 \times 10^{-5}$ |
| Length (L) | | 1 cm | 1 km | $1.0 \times 10^{-5}$ |
| Density (ρ) | | | | |
|     Overburden | | 1500 kg/m$^3$ | 2700 kg/m$^3$ | 0.55 |
|     Model salt/polymer | | 972 kg/m$^3$ | 2200 kg/m$^3$ | 0.44 |
| Density contrast | | 528 | 500 | 1.05 |
| Gravitational acceleration (g) | | 9.81 m/s$^2$ | 9.81 m/s$^2$ | 1 |
| Overburden coefficient friction (μ) | | 0.7 | 0.8 | 0.87 |
| Cohesive strength | | 55 pa | 50 x 10$^6$ Pa | 1.1 x 10$^{-5}$ |
| Deviatoric stress (σ; Pa) | $\sigma = \rho \cdot g \cdot L$ | 121 Pa | $1.17 \times 10^7$ Pa | $1.0 \times 10^{-5}$ |
| Polymer layer viscosity (η) | | $1.6 \times 10^4$ Pa·s | $10^{18}$–$10^{19}$ Pa·s | $1.6 \times 10^{-14}$–$1.6 \times 10^{-15}$ |
| Strain rate (ε) | $\varepsilon = \sigma/\eta$ | | | $3.47 \times 10^9$ |
| Time (t; s) | $t = 1/\varepsilon$ | 1 h | 0.32 Ma | |
| Velocity (v) | $v = L \cdot \varepsilon$ | 1 cm/h | 3.17 mm/y | |
| | | 0.6 cm/h | 1.90 mm/y | |

### 2.2 Experimental set-up and procedure

The experimental program consists of three models (Table 1): Model 1, as the baseline model, involves rifting of the salt-bearing segmented rift system. Model 2 comprises rifting and inversion of the rift system, and Model 3 involves rifting and inversion with syn-contractional sedimentation.

The experimental set-up uses modeling procedures similar to those reported by Dooley and Hudec (2020). The set-up was built on a metal rig with dimensions of 114 cm long by 50 cm wide and 30 cm tall. The model boundaries were laterally set by glass

walls and movable backstops at either end (Fig. 2). The basal set-up of the model consisted of a 30 cm by 50 cm latex rubber sheet located in the center of the model and attached to a 35 cm long metal base plate (with a thickness of 2 mm) on one side and a mylar plastic sheet (with a thickness <1 mm) on the other side (Fig. 2). The metal plate and mylar sheet were attached to the backstops. Extension and shortening were achieved by computer-controlled motor-driven worm-screws that pulled or pushed the movable backstops at the required velocities (see below). A 2 mm thick layer of polymer covered the basal plate. It served, first, as a basal detachment to accommodate the transference of deformation from the moving walls during extension and shortening, and second, it absorbed any undesired deformation caused by the underlying basal materials (the metal plate, rubber sheet, and mylar sheet; e.g., McClay et al., 2002; Amilibia et al., 2005; Dooley and Hudec, 2020). The model set-up does have limitations, however. Extension focused in the proximal domain with respect to both moving backstops, and as a result, marginal grabens and minibasins developed along the lateral margins of the model, close to the backstops.

Dooley and Hudec (2020) used a series of en-echelon rectangular prisms of polymer that resulted in the extensional collapse of the polymer slabs and the development of symmetrical grabens as the polymer was stretched and evacuated. This configuration, combined with the close spacing of the stepped rectangular prisms of silicone, resulted into the formation of a continuous graben in the basement and a continuous supra-salt minibasin as extensional deformation progressed without the development of significant accommodation zones (Figs. 5, 6 and 10 of Dooley and Hudec, 2020). We used triangular polymer prisms to constrain the location and development of listric faults to create half graben basin geometries, as opposed to the symmetric minibasins and basement grabens reproduced by Dooley and Hudec (2020). Both triangular prisms measured 15 cm long, 7 cm wide, and 4 cm tall. The polymer seeds were laterally and longitudinally offset from each other to promote the development of a transfer zone between the fault segments nucleated at each seed. The basal set-up and the two polymer seeds were rapidly buried to ensure the preservation of their triangular shape by 9.4 cm thick sand pack simulating the basement (i.e., white sequence in Fig. 2a). The upper part of the sand pack consisted of a layered sequence of 2 mm thick blue-white-grey sand layers to constrain deformation in the upper part of basement (Fig. 3). It is important to note that these silicone seeds do not represent preexisting salt structures, but, as stated above, localize the development of faults and half-grabens in specific positions within the model.

The different evolutionary stages of the sandbox models were indicated by a specific color sand pack (Fig. 3); multistage rift system comprised three phases in which extension was achieved by pulling both moveable walls. Extension velocities were determined from the results of several test models. The moving walls pulled the metal plate and mylar sheet attached to them and consequently stretched the rubber sheet in the center of the model, focusing extensional deformation in the overriding sand pack and stretched and thinned the silicone seeds, which localized listric faults and the development of half-grabens (Fig. 3b). In rifting stage 1, the basement sand pack was extended to create syn-rift basins above the polymer seeds and an accommodation zone. Combined velocity during this phase was 1 cm/h (i.e., the total velocity considering the movement of both moving walls). This velocity is comparable to 2.52 mm/y in nature, which falls within the range of rates observed in present-day rifts, such as the East African Rift Rift (Saria et al., 2014). Models underwent a total of 9 cm of extension during this first phase to create sufficient basement topography for the supra-salt syn-rift basins before the addition of the early syn-rift model salt layer.

At the end of the first rift stage, the entire model was covered by a layer of syn-rift model salt with a minimum thickness of 10 mm above the basement highs. The polymer was added in pieces and allowed to homogenize for 48 hours until the layer obtained a completely horizontal surface devoid of air bubbles. The model salt was covered by a regional 4 mm thick pair of white and light blue sand layers (referred to the pre-kinematic layer; Fig. 3a) simulating a pre-rift unit.

Rifting stage 2 consisted of six episodes of extension with a combined velocity rate of 0.6 cm/h (slower than stage 1) each of these followed by deposition of alternating layers of white and red sand to produce loading of the underlying syn-rift model salt. This rate is equivalent to 1.9 mm/y in nature, which while slower than the previous rate, still falls within the accepted range of values for the East African Rift (Saria et al., 2014). A slower extension rate was used in this stage because of the

strain dependent behavior of the polymer model salt layer. After the deposition of each layer, models were left for sedimentary
depocenters to downbuild, to sink into the salt forming minibasins (e.g., Rowan and Vendeville, 2006; Santolaria et al., 2021),

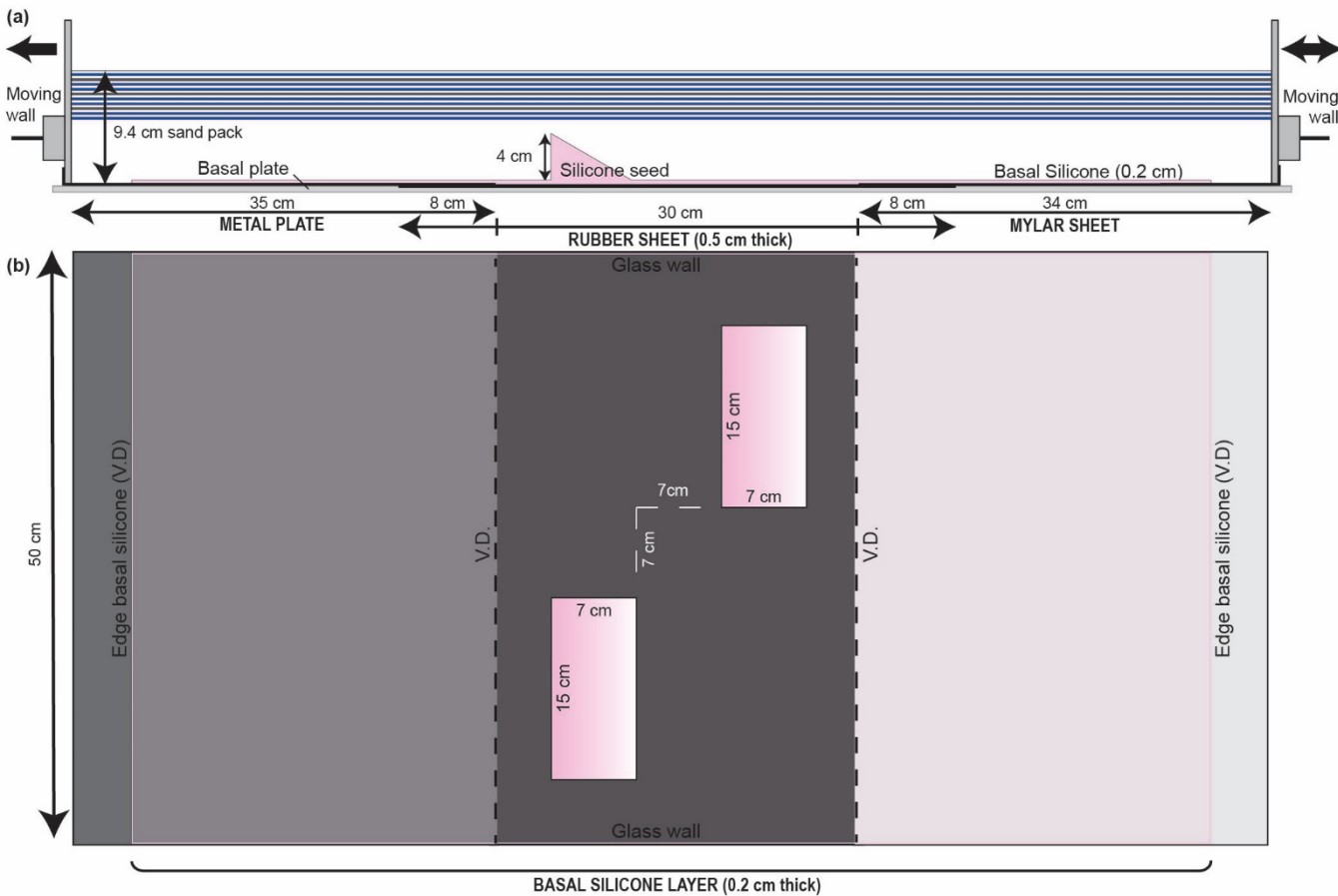

**Figure 2. Experimental set-up. (a) Section view of model set-up through one of the silicone seeds illustrating the placement of the triangular prism within the basement sand pack on top of the basal set-up. (b) The plan view figure illustrates the organization of the basal set-up materials. V.D. stands for velocity discontinuity; these discontinuities are present at the edges of the rubber sheet**
**where it connects to the metal plate and mylar sheet. The basal silicone layer that covers the basal set-up is 0.2 cm thick and extends across the entire set up. The two prismatic silicone seeds are located within the rubber sheet and are offset from each other by 7 cm in the horizontal and vertical directions.**

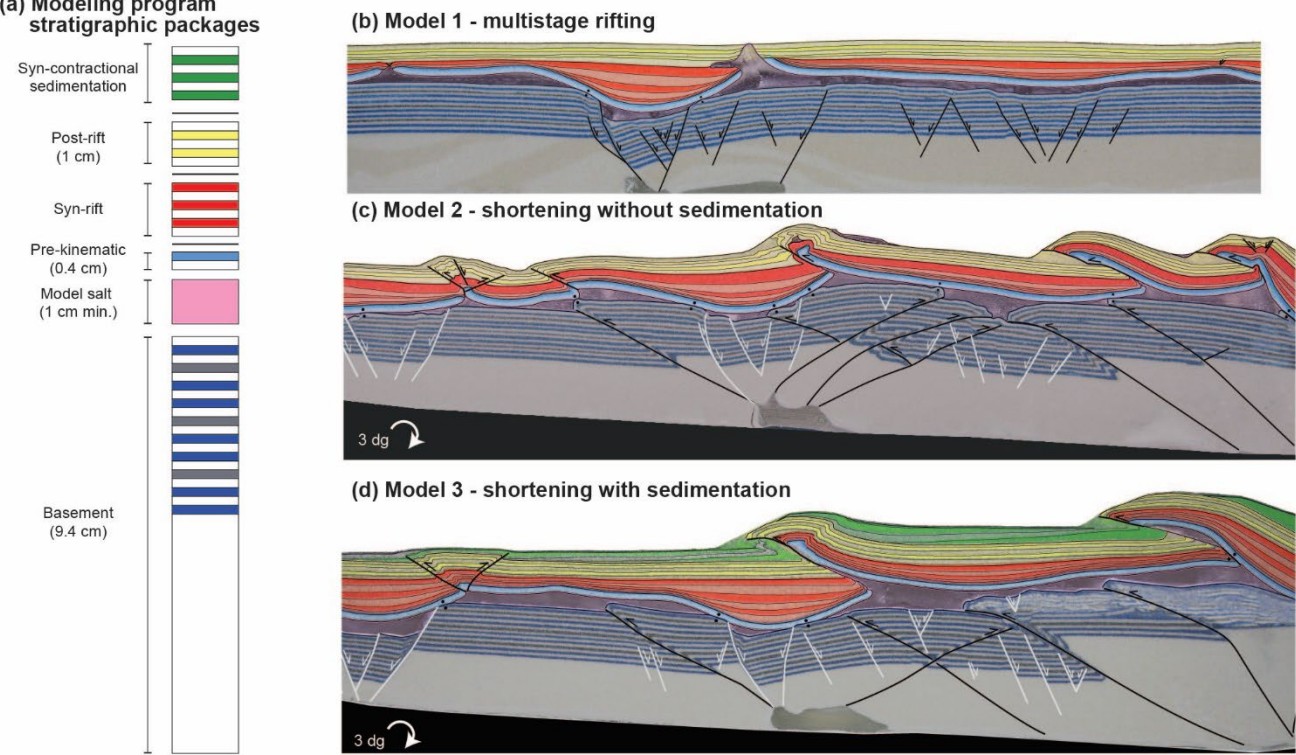

**Figure 3. Experimental tectonostratigraphic units and final distributions through each model. (a) Schematic stratigraphic column of the tectonostratigraphic sand packages and their representative colors. (b) Representative cross section through Model 1 showing the distribution of the individual sand packages of the syn-rift and post-rift succession (see Fig. 4c). (c) Representative cross section through Model 2 showing the distribution of layers at the end of shortening (see Fig. 7c). (d) Representative cross section through Model 3 at the end of shortening with the addition of syn-contractional sedimentation (see Fig. 10c).**

before the next episode of extension. For each layer, sand was added in the center of the lows created by the pulse of extension. Downbuild time for each syn-rift layer lasted from 15 h maximum, to 8 h minimum. For the first two syn-rift layers, the regional elevation of sedimentation was maintained at the top of the pre-kinematic sand pack. After that, the regional elevation of sedimentation was raised 1 mm for each syn-rift sand layer to preserve those structures formed by model salt inflation and prevent model salt extrusion to the model surface. The surface of the model was leveled using a scraper after the deposition of each sand layer to maintain the regional elevation. Leveling caused erosion on the crests of positive relief caused by salt inflation. The rate of syn-rift sedimentation increased with each layer as differential sedimentary loading increased the rate of downbuilding and extrusion of model salt from underneath the main depocenters. Any extruded model salt was periodically removed from the model surface by carefully cutting it away.

At the end of extension, the entire model was covered by a 10 mm thick post-rift succession of alternating 2 mm thick layers of white and yellow sand (post-rift phase). The layers in this succession were deposited consecutively, without any pauses for downbuilding, as in the previous extensional stage. Model 1 was stopped after the addition of the post-rift succession and used as the baseline model for the shortening experiments (Fig. 3).

Lastly, models 2 and 3 were shortened producing the inversion of the segmented extensional basins (contractional phase). Before the onset of shortening, the deformation rig was tilted 3° towards the hinterland (to the right in all section figures) to favor the model attaining its critical taper required to initiate the basal sliding of the thrust wedge with the least internal deformation possible (i.e., Chapple, 1978; Graveleau et al., 2012). Additionally, this tilt creates a low angle unconformity at

the base of the syn-contractional succession similar to that identified in the South-Central Pyrenees (e.g., Muñoz et al., 2018). After the extension and before the onset of shortening, the mylar was detached from the moving wall and fixed to the basal plate to detach and shorten the sand by moving the backstop wall over the mylar sheet. Shortening was accomplished by moving the wall on the righthand side of the model, previously attached to the mylar sheet (i.e., push from behind model

experimental design) to a maximum of 25 cm of shortening with a velocity of 0.6 cm/h, which is equivalent to 25 km and 1.9 mm/y in nature, respectively. This phase of the model differs from the Dooley and Hudec (2020) set-up which did not involve tilting of the deformation rig and used dual motor shortening by which both moving walls compressed the model, preventing the propagation of a thrust front. Model 2 was shortened without erosion or addition of syn-contractional sand. Conversely, Model 3 was shortened with the addition of alternating green and white syn-contractional sand layers. Syn-contractional sand

layers were added every 3.6 cm of shortening (sedimentary rate of 2 mm/6 hours, or 0.08 mm/y in nature, at the toe of the system), raising the regional 2 mm for each new layer. During shortening, extruded model salt was removed periodically from the surface simulating erosion.

## 2.3 Data capture and analytical techniques

Overhead high-resolution time-lapse photographs were taken every 90 seconds by a computer-controlled digital camera to

record the kinematic evolution of the experiments. Videos from the time-lapse photos of each model have been generated (see Supplementary Material). To simplify the description of the experimental results, the cover structures have been named according to their time of development and structural style (i.e., T1, T2, BT3, etc.; T for thrusts, BT for backthrusts, and P for pop-up structures, G for grabens).

At the end of each experiment, the models were sliced at intervals of 3 mm using an apparatus designed by the Geomodels

laboratory and photographed. Photos taken of the serial vertical cross-sections were then imported to image-processing software to produce 3D voxel models by interpolating the space between each cross-sectional image to create a 3D volume. These voxels were used to extract virtual sections such as inlines and depth-slices for analyzing the 3D structural variation of each model (Dooley et al., 2005; Ferrer et al., 2016; Granado et al., 2017; Santolaria et al., 2021). Voxels were also used to interpret key stratigraphic horizons, fault planes, welds and to extract structure maps and true stratigraphic thickness maps of

key sand packages.

## 3. Experimental Results

### 3.1 Model 1: Multistage Rifting

This section describes the experimental results of the baseline Model 1 to illustrate the 3D geometries, tectonostratigraphic architecture, and the structural styles of supra-salt rift basins that developed above the segmented half-grabens separated by an

extensional accommodation zone that developed during the basement-involved extensional phases. Serial sections of this model from the final slicing of the model and the virtual sections produced from those highlight the 3D geometries, tectonostratigraphic architecture, and the structural styles developed during the basement-involved extensional phases (Fig. 4). These sections highlight the lateral termination of the basement faults along strike, the location and extent of model salt structures, and the changes in thickness of the model salt, syn-rift succession, and post-rift succession. Top view photographs

(Fig. 5) and the time-lapse video (see Supplementary Material for Model 1) illustrate the evolution of the model surface through the second phase of extension, as well as the erosional leveling of the cover sequence and the inflation and extrusion of model salt. Selected structure and true stratigraphic thickness maps show the final 3D distribution of the basement fault system, the model salt, and the syn-rift succession at the end of the model (Fig. 6).

Cross-sections show master basement faults with a concave upwards listric geometry rooted at the highest point of the polymer

seeds (Master Fault 1 and Master Fault 2 in Figs. 4a, c). These faults trend orthogonal to the extension direction, while their

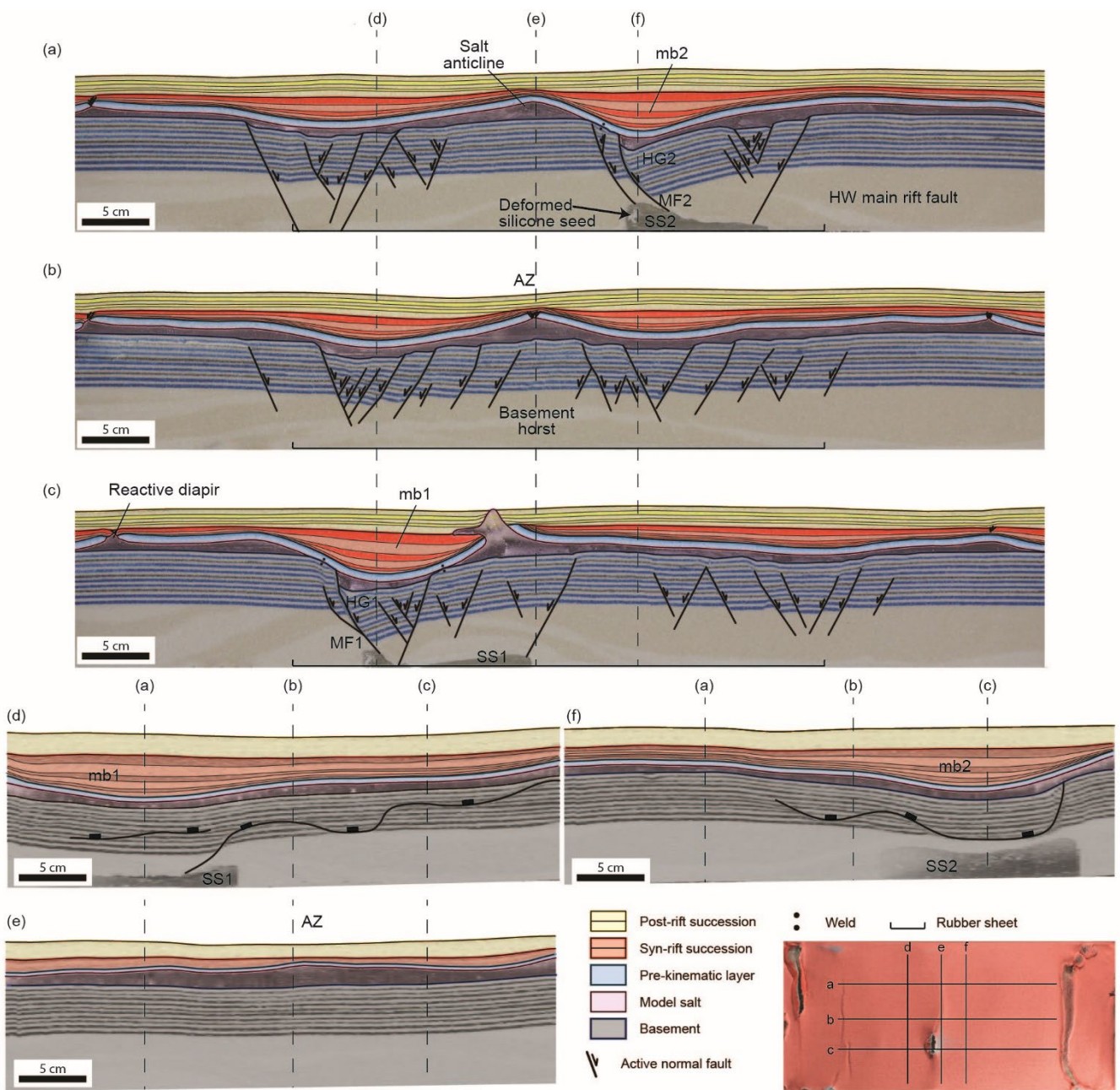

**Figure 4. Interpreted representative cross sections (a-c), and virtual inlines (d-f) through the central minibasins (mb1 and mb2) and the basement accommodation zone of the results of Model 1 (multistage rift model). Inset top view image shows section locations. Abbreviations: HG – half graben; HW – hanging wall.; mb – minibasin; MF – Master Fault; SS – silicone seed; AZ – accommodation zone.**

lateral extension is imposed by the width of the polymer seeds and do not extend across the accommodation zone (Fig. 4b). The triangular shape of both seeds constrains the development of another listric fault in the footwall of the master ones. Slip attained by the master faults produces half-grabens above both polymer seeds, characterized by hanging wall rollover

anticlines. In the case of Master Fault 2, this rollover has an associated crestal collapse graben. The rollover anticlines are also bound by antithetic extensional faults developed at the other tip of the polymer seeds (Fig. 4a, c). Master Fault 1 attains more slip than Master Fault 2, and consequently, both faults create different accommodation space at the syncline minibasins developed on their downthrown hanging walls. Conversely, the intervening accommodation zone in between Master Fault 1 and Master Fault 2 is characterized by a dense array of left- and right-dipping extensional faults with limited displacement describing a central horst bounded by two grabens (Fig. 4b).

While the structural style of basement is characterized by faulting and rollover anticlines, the salt overburden is characterized by decoupled deformation in the form of folding (i.e., broad synclines and anticlines) and salt structures (Fig. 4a to c). Synclinal basins develop above the hanging walls of Master Fault 1 and Master Fault 2 (minibasin 1 and minibasin 2, respectively). These synclinal depocenters are filled by syn-extensional layers that pinch out laterally. The model salt layer shows important thickness differences, being totally depleted above the major basement faults to thickened by inflation in the footwall of these faults and above the basement rollover hinges (Fig. 4 a and c). A diapir developed by crestal extension and erosion of the salt inflated anticline developed above the rollover anticline of Master Fault 1, allowing the extrusion of a salt glacier (Fig. 4c). In the accommodation zone, inflated salt above the basement horst separates the lateral terminations of both synclinal minibasins (Fig. 4b).

Virtual inlines (Fig. 4 d to f) show the geometry of the syn-rift minibasins along strike, the distribution of model salt, and the basement fault plane intersections. In these sections, the syn-rift minibasins display similar geometries to those displayed in the cross-sections, with laterally lapping early syn-rift layers, and their thickest succession coincident with the center of the minibasin. The distribution of model salt is roughly even across the inlines passing through the minibasins depocenters. By contrast, model salt is thickest and shows an uneven distribution through the transfer zone, where the syn-rift succession is thinnest (Fig. 4e).

Early thick-skinned extension after deposition of the pre-kinematic layer triggered model salt movement and formation of drape folds above the major basement faults, while salt inflation occurred at the basement rollover hinge (Fig. 5a). Some grabens developed along the crest of developing salt anticlines near the edges of the model (Fig. 5a). As extension increased, migration of model salt from below the synclines towards the hinge of the basement rollovers increased producing significant inflation in salt-cored anticlines (blueish colors in Fig. 5b). At the same time, drape monoclines formed above major basement faults also allowing the development of salt-inflated anticlines in the footwall of those faults (Figs. 4a, 5b, and 6c). The stretching produced at the crest of some of these anticlines allow the development of grabens that thinned the overburden (i.e., minibasin 1, Fig. 5b). In addition, the relief generated by salt inflation favored the erosion of many of these anticlines (blueish colors in Fig. 5b). After 4.2 cm of extension, the syncline basins propagate laterally. At this point a depressed zone separated the anticline developed at the hinge of the basement rollover related to minibasin 1 and the one developed above the footwall of Master Fault 2 (Fig. 5b). As deformation progressed, the syncline basins become wider and gentler (Fig. 5c) as the last broadly uniform syn-rift layers indicate (Fig. 4a-c). The sedimentation rate applied to the model resulted in the burial of most of the salt-cored anticlines, and a diapir only developed during this late extensional stage (Fig. 5c). This diapir formed by a combination of crestal stretching and erosion as the unconformity between pre-kinematic and syn-kinematic layers indicates at both flanks of the structure (Fig. 4c). A salt sheet extruding towards the inner part of the syncline basin formed during the end of rifting (Fig. 5c) and the diapir grew passively during the deposition of the post-rift sand pack (Fig. 4c).

Thickness of the post-rift layers is quite uniform. However, close to the model salt sheet (Fig. 5c), thickness of the first post-rift layer increases towards the extruding model salt, below which the base of minibasin 1 is welded to the top of the basement (Fig. 4c). Timing of welding is thus coincident with the end of rifting for the footwall welds, while welding of minibasin 1 below the thickest post-rift succession occurs during post-rift salt extrusion associated with sediment loading.

The top basement structure map shows two main structural lows associated with the displacement maxima of the master faults above the silicone seeds (Fig. 6a). Half-graben 1 propagated along strike across the whole model width (Fig. 4 a-c). This is probably related to the presence of the underlying velocity discontinuity (V.D. in Fig. 2) that favors extension localization,

lateral slip transfer along the strike of the silicone seed, and the formation of the largest depocenter of Model 1. Conversely, extension along Master Fault 2 produces a more diffuse structural pattern, with one largest depocenter right of the master fault, and another depocenter shifted to the bottom right of the model, over the velocity discontinuity (Fig. 6a). As shown by the structural maps (Fig. 6a), the basement top at the accommodation zone lays deeper than the basement highs.

A true thickness map of the autochthonous model salt (i.e., true stratigraphic thickness map in Fig. 6b) depicts the position and shape of the welded areas of minibasins 1 and 2 ("W" in Fig. 6b). The map also shows thinning and welding of the autochthonous salt at the tip of the footwall blocks of the extensional faults (i.e., nearly welded areas; compare Fig. 4 and Fig. 6d). An apparent single salt anticline continues from the hanging wall of Master Fault 1 links with the salt anticline in the footwall of Master Fault 2 across the accommodation zone (Figs. 6c and d).

Distribution of the syn-rift succession thickness is illustrated on the true stratigraphic thickness map (i.e., true stratigraphic thickness map on Fig. 6c). The map shows that the syn-rift depocenters display an elliptical pattern, whose thickness distribution follows the location and trend of the basement extensional faults. A ridge of thin syn-rift sediments links the footwall of Master Fault 2 with the hanging wall of Master Fault 1, marking the trend of a salt anticline across the accommodation zone. Towards Master Fault 1 hanging wall the salt anticline becomes a salt-extruding wall (Fig. 6c). Additional syn-rift depocenters also follow the trend of the basement top and occur shifted from the position of minibasin 1 and minibasin 2. These depocenters are proximal to the velocity discontinuities (i.e., edges of the rubber sheet).

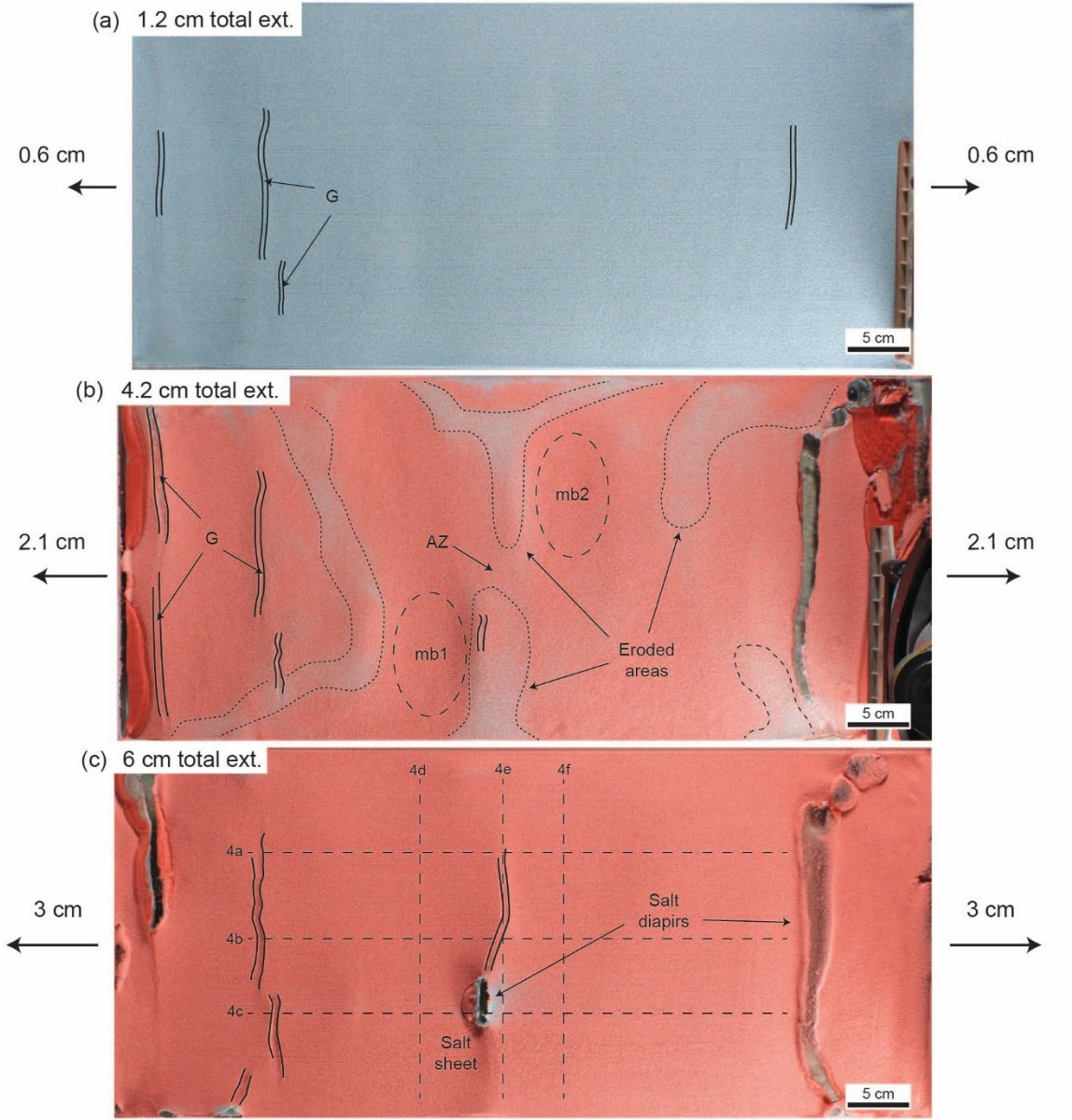


**Figure 5. Overhead views illustrating the surface evolution of Model 1 (multistage rifting) after 1.2, 4.2 and 6 cm of extension, respectively. (a) Incipient grabens developed during the early extension (1.2 cm) at the top of the pre-kinematic overburden; (b) syncline basins (minibasin 1 and minibasin 2) development constrained by the slip of basement faults rooted at the polymer seeds during mild extension. Salt-inflated areas (blue-reddish color) are locally eroded; and (c) a salt sheet extrudes from a passive diapir developed by erosional piercement during the late extension. During this stage, synclinal basins are wider. Abbreviations: ext. – extension; G – graben; mb – minibasin; AZ – accommodation zone.**


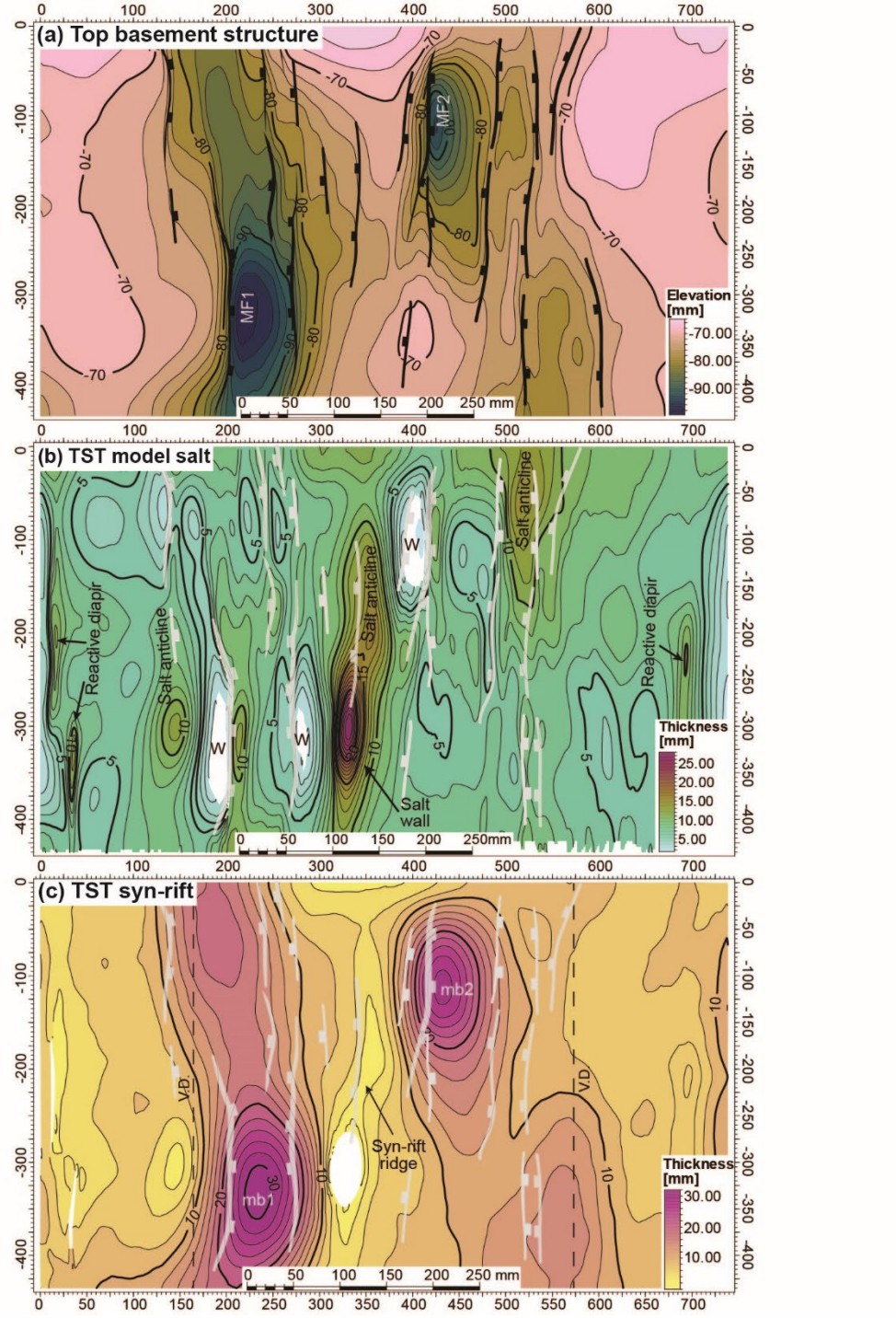

**Figure 6. Structure and true stratigraphic thickness (TST) maps of key horizons from the voxel results of Model 1 (multistage rifting): (a) Top basement structure map, (b) TST autochthonous salt map, and (c) TST syn-rift map. Light gray lines in figures b**

**and c represent the basement structures. Abbreviations: mb – minibasin; MF – Master Fault; V.D. – velocity discontinuity; W – weld.**

### 3.2 Model 2: Shortening without syn-contractional sedimentation

This section describes the results of Model 2, which was subjected to 25 cm of shortening after undergoing the same extensional phases as Model 1 (Fig. 7). Besides the periodic removal of extruded silicone, no addition of syn-contractional surface
processes (i.e., erosion or sedimentation) were performed on the model. The serial sections highlight the 3D geometries of deformed salt structures and the involvement of the pre-salt basement in a broadly foreland propagating sequence of thrust imbrication (Figs. 7a-d). Top view photographs (Fig. 8) and time lapse videos (see Supplementary Material for Model 2) illustrate the kinematics of the salt-detached thrust system, and the contractional deformation of salt structures developed during previous extensional phases. In both the sections and top view photographs, the structures in the cover are labeled by a
letter representing structure type followed by a number representing the order in which the structures developed. Degradation of the frontal limbs of surface-breaching thrust-related folds is common throughout the modeling.

During shortening the contractional deformation has also been decoupled along the syn-rift model salt. Underneath, the basement-involved thrust system has been detached along the basal silicone layer truncating the previously developed extensional faults. However, the master basement extensional faults have been partially reactivated along the shallow dipping
portions, as the main thrusts have also been nucleated in the silicone seeds, thus reproducing inversion tectonics geometries, such as short-cuts (Fig 7. a-d). The distribution of the model surface elevation relates to the inversion of these basement extensional faults. Thus, the lowest regional elevation is located in the footwall of the thrusts inverting the master extensional faults (Fig 7. A, c). The section through minibasin 2 displays a broad hinterland plateau in the hanging wall of the inverted Master Fault 2, which shows a short-cut in the upper part of extensional Master Fault 2 (Fig. 7a). The hinterland pre-salt
basement is also affected by two foreland propagating thrusts which transfer the displacement upwards into the syn-rift model salt; late backthrusts cut these thrusts. The contractional structure of the post-salt succession is represented by salt-detached thrusts and backthrusts and related folds (Fig. 7a). Unlike the basement structure, there is not a predominant vergence. After 25 cm of shortening minibasin 2 remains welded to the basement at two points, the footwall and the hanging wall of the inverted Master Fault 2 basement fault (i.e., its original location according to Model 1). Above, the thinner syn-rift cover
sequence deposited on the footwall of the Master Fault 2 is backthrusted over the syn-rift depocenter of minibasin 2, generating the highest structural elevation of the section. Additional structures are: a pop-up in the foreland related to the squeezing and secondary welding of diapirs (P4); a salt-detached backthrust transporting minibasin 2 with a frontal pop-up structure (BT3); a hinterland-dipping thrust involving the marginal syn-rift minibasin (T1); triangle zone welded to the basement buried below resedimented post-rift sediments and allochthonous salt sheet (Fig. 7a).

The section along the basement accommodation zone is characterized by a smoother change in the topography in comparison to the previous section (Fig. 7b). The basement structure corresponds to an imbricated wedge of thrusts and backthrusts, cutting through inherited extensional faults. The section also displays two structural highs related to salt-detached structures. The regional elevation is controlled by the basement thrust system, however, the topographic surface ahead of the frontal thrust system is more gradual than in the previous section (Fig. 7a, b). The detached cover succession in the foreland is characterized
by two backthrusts, one formed after the squeezing of a salt wall (BT4) and a backthrust located immediately ahead of the foremost basement thrust tip (BT6). The central salt structure is formed by a fully welded salt wall with a preserved pedestal, flanked by the lateral terminations of syn-rift minibasin 1 and minibasin 2. A salt sheet developed above this structure, with the majority of the salt extrusion flowing towards the hinterland in response to the hinterland-dipping slope. Time of salt extrusion began during the post-rift, as shown by the allochthonous salt deposited on top of the syn-rift succession, and
continued to extrude during shortening, until the salt sheet was cut off from its source due to welding. At the hinterland, thrusts and a backthrust interact each other to develop pop-ups and a triangle zone underneath an allochthonous salt sheet (Fig. 7b).

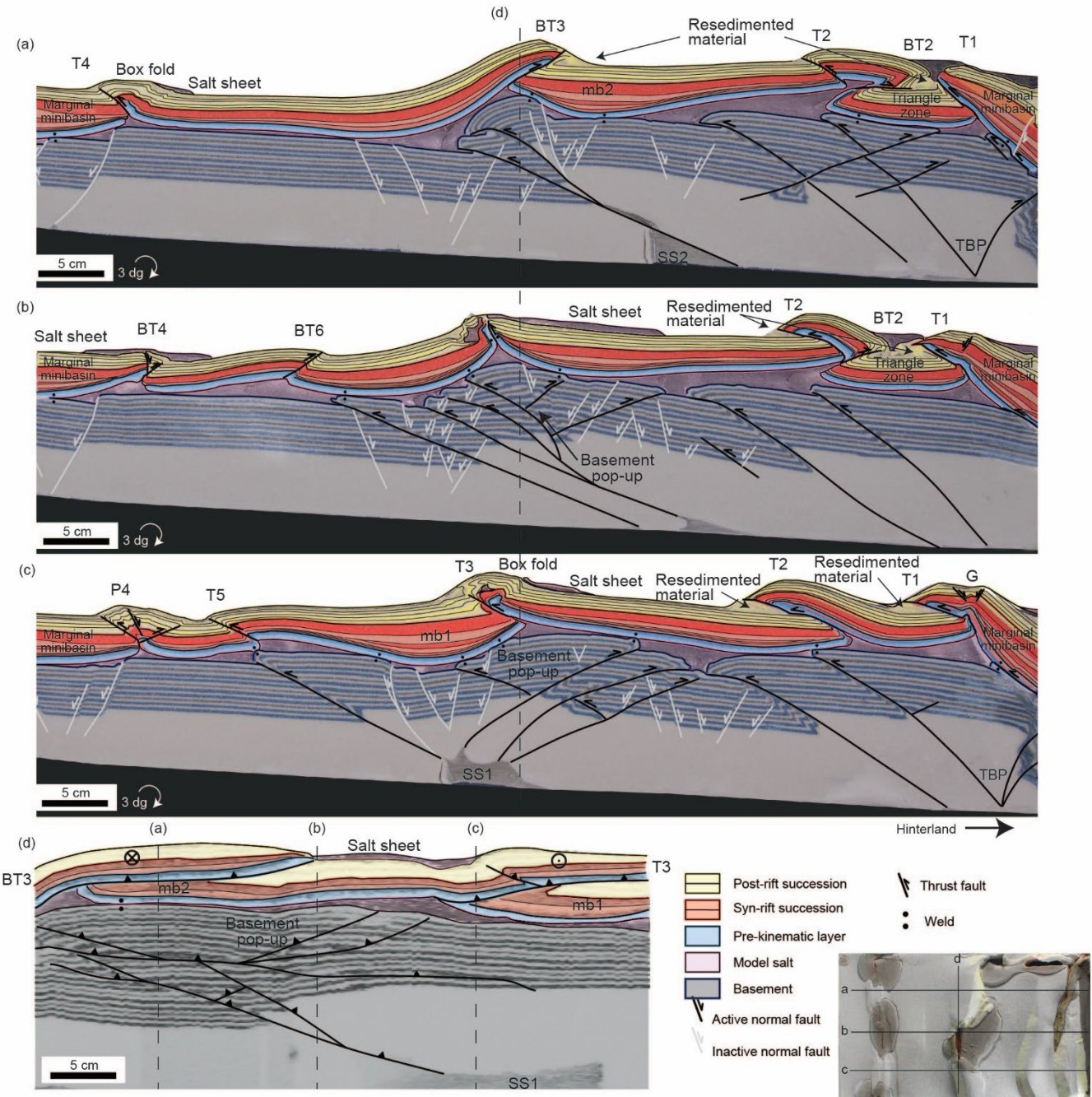

**Figure 7. Interpreted representative cross sections (a-c) through the central minibasins (mb1 and mb2) and the basement accommodation zone of the results of Model 2 (shortening, no sedimentation), plus a virtual inline through the accommodation zone (d). Individual structures in the cover thrust system are labelled according to their timing of kinematic evolution. Inset top view image shows section locations. Abbreviations: BT – backthrust; G – graben; mb – minibasin; P – pop-up structure; SS – silicone seed; T – thrust; TBP – trailing branch point.**


The section through minibasin 1 shows a different topographic distribution than the previous two sections (Fig. 7c). The frontal basement thrust is located further to the left (foreland) because of the inversion of Master Fault 1 and its silicone seed (Silicone Seed 2). This seed also controlled the nucleation of a backthrust system which produced the uplift of the former rollover anticline and the highest structural elevation. The post-salt succession displays a consistent foreland vergence of thrusts and related folds, contrary to the other two sections of Model 2. Minibasin 1 preserved a relatively low structural position at the end of the contractional deformation in the footwall of the thrust that resulted from the break-through of the squeezed salt anticline and related salt structure. In addition, minibasin 1 is welded to the basement at three different points – on the footwall of Master Fault 1, and in two places on the hanging wall. Foreland verging thrusts developed in the hinterland part of the model. The marginal minibasin at the rear of the wedge is imbricated on a thrust sheet and folded into two dipping panels: a shallowly-dipping thrust tip, and steeply dipping back-limb. The transition of both panels is marked by a crestal collapse graben structure (Fig. 7c).

A virtual inline through the basement accommodation zone shows the lateral stack of the post-salt succession with the reduced syn-rift succession in a hanging wall position of either the thrust or the backthrust (Fig. 7a, c, d). Model salt thickness is thinnest below minibasin 1 and minibasin 2 (even welded below minibasin 2) and thickens in the basement accommodation zone with the thickest position along the margin of minibasin 1.

The kinematic evolution of the cover sequence during the shortening phase can be tracked using top view images, thrust chronology chart, and time-lapse videos (see Supplementary Material for Model 2, and Fig. 8). The post-salt thrust system developed as a foreland propagating (right to left) sequence, although salt structures were reactivated in the foreland ahead of the thrust front since early stages of shortening (Fig. 8). At the onset of shortening, extensional faults and diapirs occurred in the left margin (foreland) of the model (Fig. 8a). This extensional deformation probably developed in response to the 3° tilt of the deformation rig toward the hinterland. The first thrust (T1) developed in the hinterland against the backstop (Fig. 8a). Time-lapse videos show that salt wall squeezing occurred across the model after about 6 cm of shortening (~25% of total shortening) coeval with the development of the second foreland verging thrust in the hinterland, Thrust 2 (Fig 8b and Supplementary Material). The backthrust (BT2) to the right of Thrust 2 as can be seen in sections 7a and 7b, are not clearly visible from the top view images, except for a slight indenture in the hanging wall of Thrust 2 (Figs. 8b-d). After 12 cm of shortening (~50% of total shortening) thrusts and backthrusts nucleate along the salt walls in the model (Fig. 8c). The thrust chronology chart displays an irregular timing of thrust nucleation with irregular time gaps between the initiation of each thrust (Fig. 8e).

The accommodation zone between the master extensional faults at the onset of shortening is characterized by the presence of a diapir located at the intersection between the salt anticline developed above the rollover anticline of Master Fault 1 and the salt anticline at the footwall of Master Fault 2 (compare Figs. 5, 6 and 8). This diapir has been squeezed since early stages of shortening and feeds an allochthonous salt sheet that occupies the structural low between the inverted minibasin 1 and minibasin 2 minibasins (Figs. 7d and 8). As deformation progressed, the backthrust that resulted from the squeezing of the salt anticline in the footwall of Master Fault 2 (BT3) and the thrust at the Master Fault 1 rollover salt anticline (T3) connect with the salt weld of the squeezed diapir (Figs. 7 and 8). Near the end of shortening, a thrust (T5) and backthrust (BT6) develop in the foreland ahead of the accommodation zone (Fig. 8c).

The top basement structure and true stratigraphic thickness maps of the syn-rift succession and model salt are shown in Figure 9. The top basement structure map shows foreland propagating thrusts. The highest elevation of the basement sits in the center of the model and is the result of the connection of the short-cut of the inverted Master Fault 2 fault with the pop-up developed on the previous rollover anticline of Master Fault 1 fault. A subtle change of strike and a decrease of structural relief along this high occur in the accommodation zone (Fig. 9a). The basement-involved frontal thrust shows a significant change of strike and a resulting concave geometry in map view as a result of the connection of the inverted major extensional faults forward of the accommodation zone, contrasting with the more prominent uniform structural trend in the supra-salt succession (Fig. 9b).

The model salt distribution map shows the relationship between the model salt and both the basement and cover thrust systems (Fig. 9b). The salt thickness map reveals that the number of welds increases two-fold in this shortened model. The preexisting primary welds along minibasin 1 and minibasin 2 are more extensive than at the end rifting (Model 1), forming more elongate and wider zones of contact between the cover and basement systems (Figs. 6b, 9b). The preexisting salt walls in the accommodation zone and foreland of Model 2 act as sources for salt sheets until they are secondary welded (Fig. 8a-d) and now appear as high amplitude and short wavelength anticlines adjacent to elongated secondary thrust welds (Fig. 9b). The distribution of model salt is influenced by the geometry of the basement and cover thrust systems. The basement thrust fronts can easily be traced in the true stratigraphic thickness map and the cover thrusts initiate on the areas of early model salt structures (Figs. 9b, 7a-c).


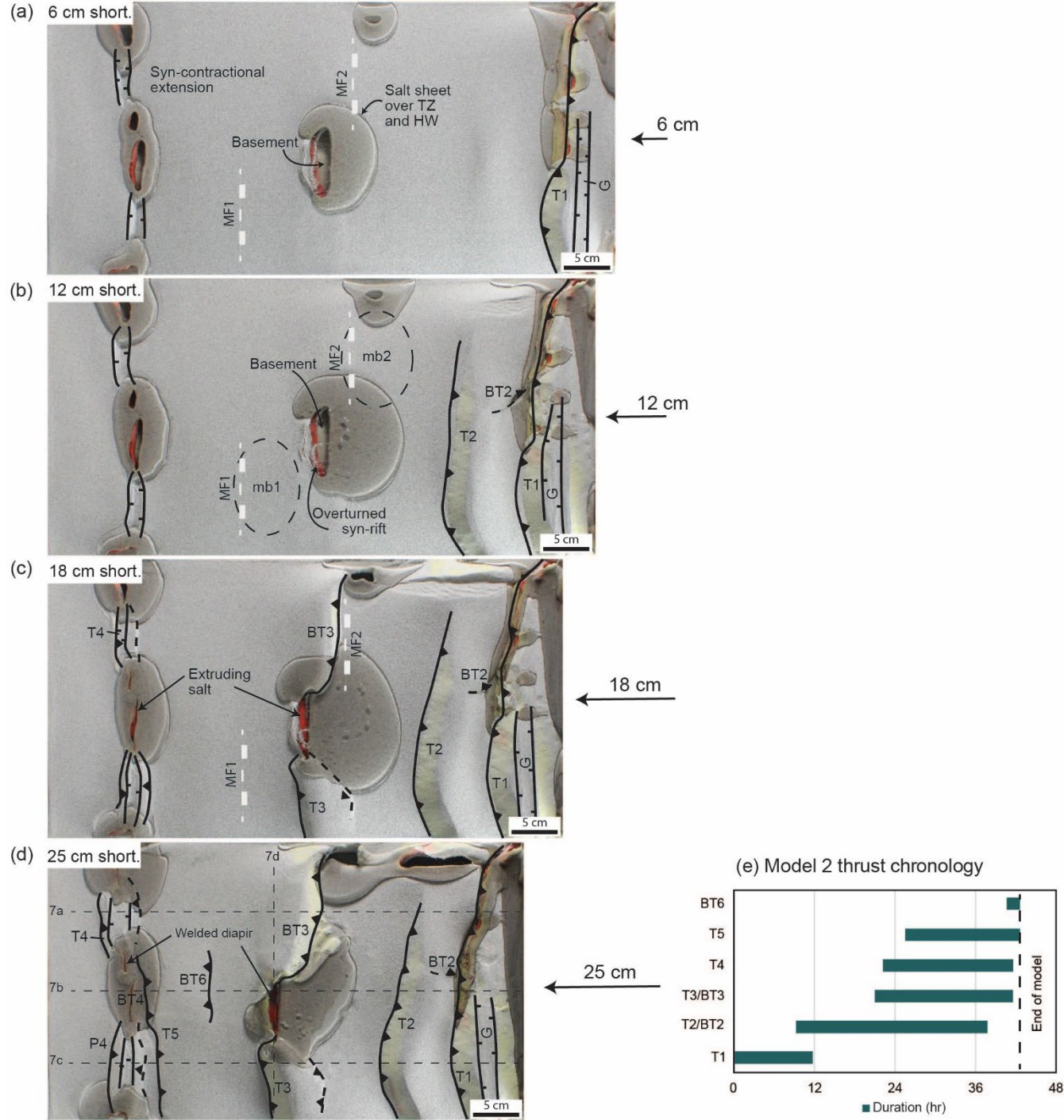


**Figure 8. (a-d)** Top-view images illustrating the contractional kinematic evolution of Model 2 (shortening, no sedimentation), including the timing of thrust initiation, model salt extrusion from salt sheets, contractional welding of salt sheets. **(e)** Graph illustrating the thrust chronology of the major thrusts and backthrusts. Abbreviations: BT – backthrust; G – graben; mb – minibasin; MF – Master Fault; P – pop-up structure; short. – shortening; T – thrust.

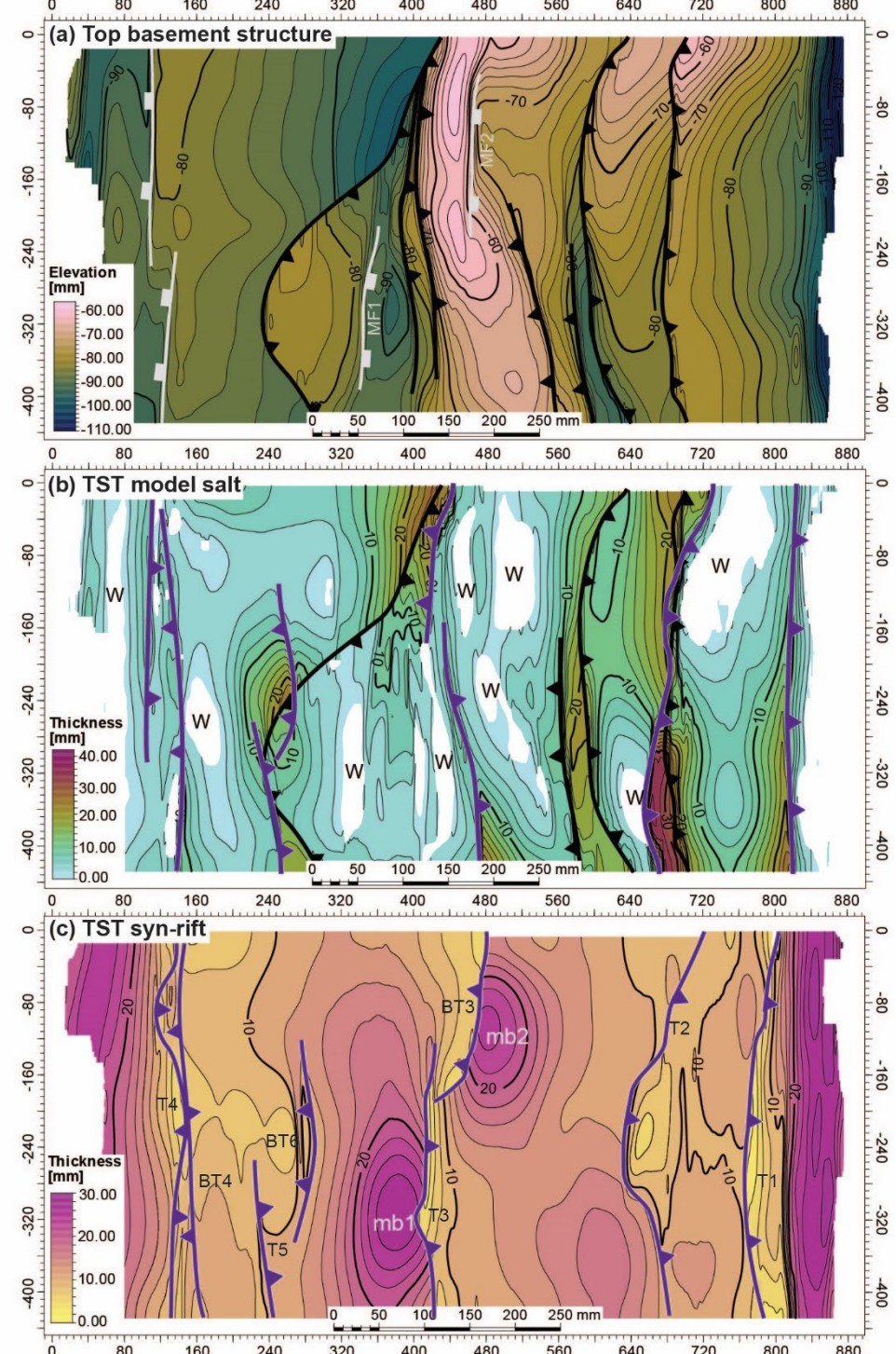


**Figure 9. Structure and true stratigraphic thickness (TST) maps of key horizons from the voxel results of Model 2 (shortening, no sedimentation): (a) Top basement structure map, (b) TST autochthonous salt map, (c) TST syn-rift map. Basement thrusts are in**

The true stratigraphic thickness syn-rift map highlights the distribution of the syn-rift succession and location of the cover thrust systems in relation to the syn-rift depocenters (Fig. 9c). Across the model, the syn-rift depocenters remain in the footwalls of the thrusts, except for the marginal minibasin in the hanging wall of Thrust 1 which is adjacent to the moving backstop and forms the first cover thrust (T1).

### 3.3 Model 3: Shortening with syn-contractional sedimentation

Model 3 was subjected to the same extensional and shortening processes as Model 2. In addition to the periodic removal of extruded silicone, syn-contractional sedimentation was added every 6 hours (every 3.6 cm of shortening), while the regional was raised 2 mm for each layer. Serial sections highlight the foreland progradation of a syn-contractional sedimentary wedge that thins towards the foreland, onlapping the underlying salt-detached structures in the cover sequence (Fig. 10). Top view photographs (Fig. 11) and time-lapse videos (see Supplementary Material for Model 3) illustrate the influence of the syn-
contractional sedimentation on the kinematics of the salt-detached thrust system, and the contractional deformation of preexisting salt structures. In both the sections and top view photographs, the structures in the cover are labeled with a letter representing structure type followed by a number representing the order in which the structures developed. As in Model 2, degradation of the frontal limbs of surface-breaching thrust-related folds is common throughout.

    As in Model 2, the master extensional faults have been reactivated by the nucleation of the thrusts on the silicone seeds. The
basement thrust system detaches on the model salt layer, crosscut the minor extensional faults, and control the slope of the model surface (Fig. 10a-c). The surface slope consistently dips to the left across all three sections. In the section through minibasin 2, the frontal thrust reactivating Master Fault 2 controlled the location of the syn-contractional depocenter in its footwall and the uplift of minibasin 2 (Fig. 10a). The accommodation space of this depocenter increased by salt evacuation of the former salt inflated area in the footwall of the preexisting Master Fault 2 extensional fault (compare Figs. 4a and 10a).
Evacuated salt was extruded to form a salt sheet forward of minibasin 2. Uplift of minibasin 2 in the hanging wall of the thrust reduced the accommodation space for the syn-contractional succession deposited above, which contrasts with the thick syn-contractional succession above the thin syn-rift series in the footwall of the thrust, thus reproducing a classical basin inversion geometry, although no extensional faults affected the post-salt succession (Figs. 4a and 10a). The post-salt cover sequence has been decoupled from the basement regardless of whether it has been welded to the basement in multiple places. The foreland
propagating thrusts in the cover sequence are sourced by model salt, which has broken through the thrust tips and extrudes on to the surface as allochthonous salt sheets (see Supplementary Material). The footwall and hanging wall cutoffs for thrusts Thrust 1 and Thrust 3, as well as the thrust welded salt wall, are characterized by reduced stratigraphy of the syn-rift and post-rift successions, demonstrating the presence of inflated model salt in these positions prior to shortening. The thrust tip of T1 is overturned and folded over the underlying syn-contractional sedimentary wedge. The salt wall that developed to the right of
minibasin 2 during extension is secondary thrust welded at the pre-kinematic layer while model salt is preserved in the salt stock above the weld (Fig. 10a). A backthrust (BT4) developed in the foreland cover sequence to the right of the welded marginal minibasin. Across the section, the syn-rift minibasins are incorporated in the hanging walls of the thrusts and backthrust.

    The section through the basement accommodation zone reveals a cover thrust system decoupled from the basement thrust
stack. In only one point at the foreland the marginal syn-rift minibasin is welded to the footwall of a preexisting normal fault (Fig. 10b). A significant amount of model salt is present in the section above the basement accommodation zone. The basement thrust stack through the accommodation zone is characterized by foreland propagating thrusts, unlike the large imbricate thrust fan that occupies the accommodation zone in Model 2 (Figs. 7b, 10b).

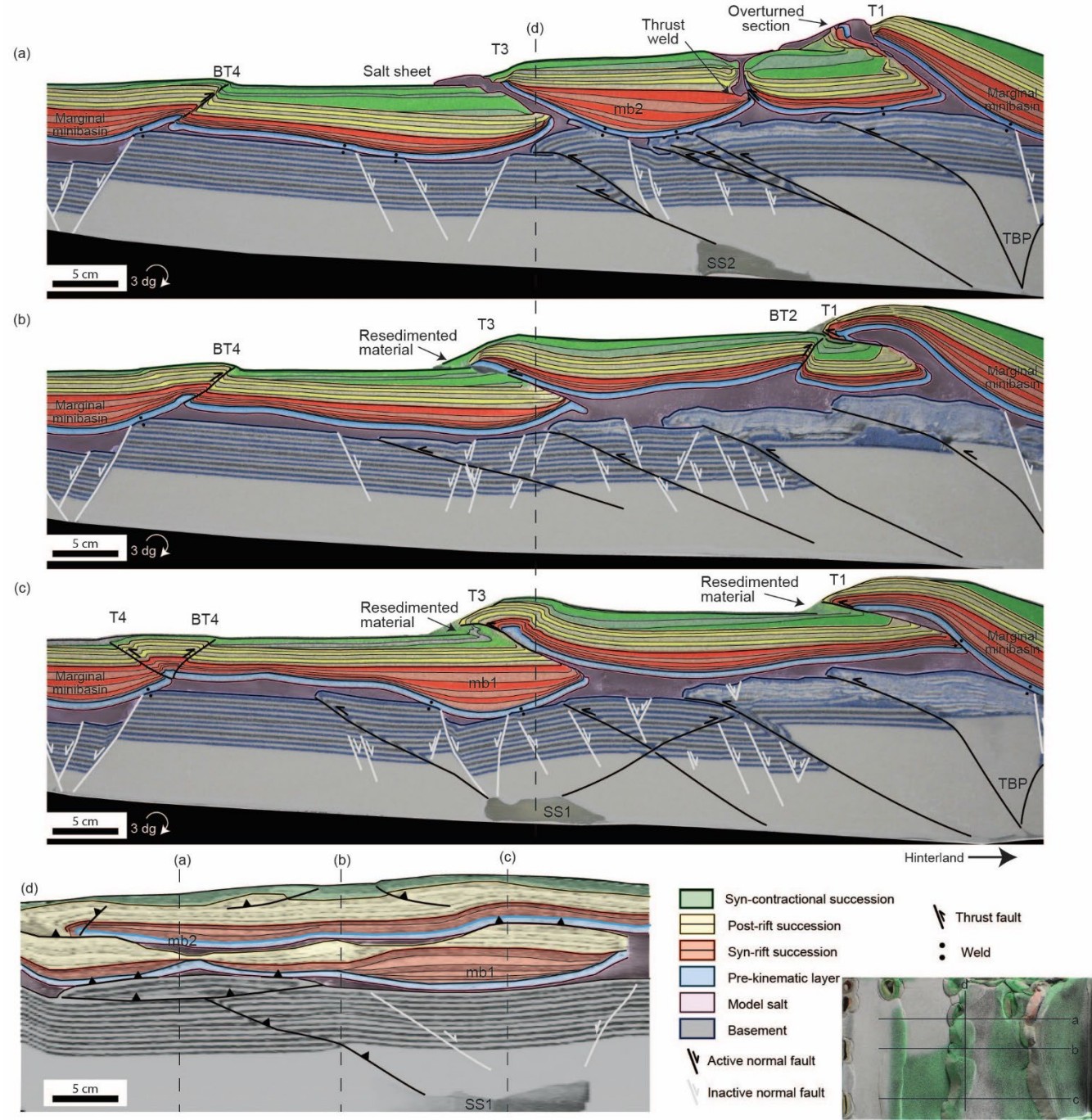

Figure 10. Interpreted representative cross sections (a-c) ad virtual inlines (d) through the central minibasins (mb1 and mb2) and the basement accommodation zone of the results of Mode 3 (shortening with syn-contractional sedimentation). Individual structures in the cover thrust system are labelled according to their timing of kinematic evolution. Inset top view image shows section locations. Abbreviations: BT – backthrust; mb – minibasin; P – pop-up; SS – silicone seed; T – thrust; TBP – trailing branch point,).

Along the section through minibasin 1, the basement thrust system extends farther into the foreland by reactivation of the silicone seed related with Master Fault 1 (Fig. 10c). Little shortening is taken up by the two thrusts and backthrust that developed in the central area of the section, while the two thrusts that nucleate in the trailing branch point (TBP) accommodate most of the shortening in the basement across this section. Unlike minibasin 2, minibasin 1 has not experienced uplift in the hanging wall of thrust that reactivated Master Fault 1, and the depocenter of the syn-contractional succession is located above the minibasin depocenter. The existence of the basement frontal thrust cannot be inferred from the geometry of the post-salt succession (Fig. 10c). The cover sequence welds to the basement in four locations; one weld per marginal minibasin and two welds along minibasin 1, one weld on to the footwall of Master Fault 1 and the other weld on its hanging wall (Fig. 10c). Less model salt is preserved in the cover thrusts in comparison to the other sections across Model 3. The hanging wall of Thrust 1 is thrust welded on the lower reaches of its footwall, preserving inflated model salt below the central part of its hanging wall. Thrust 3 has no model salt along its base, and the thrust tip is characterized by a pop-up structure above an overturned and folded syn-contractional footwall syncline. The hanging wall of Thrust 3 forms a very shallowly dipping panel towards the hinterland, whereas Backthrust 3 is not present in this section. Minibasin 1 is welded to the basement in the footwall of Thrust 3, and on the footwall of Master Fault 1. The foreland-most cover structure in this section developed on the hinterland side of the welded marginal minibasin as well and corresponds to a pop-up structure.

A virtual inline from the 3D seismic volume generated from the sliced sections shows along-strike changes of the geometry of the repeated cover succession (Fig. 10d). The highest basement elevation sits on the left, below minibasin 2, where the cover sequence is welded in two places and does not coincide with the highest structural elevation. Across the inline, the model salt is discontinuous, interrupted by welds at the lower source layer, with a thin allochthonous layer in the central part of the line between the footwall and hanging wall of Thrust 3. The syn-contractional sedimentary wedge thins over the crests of the uppermost folds near thrust tips and is thickest in the center of the line.

The kinematic evolution of the cover sequence during the shortening phase can be tracked using time-lapse videos and is illustrated here by top view images and a thrust chronology chart (see Supplementary Material and Fig. 11). The post-salt thrust system developed as a foreland propagating (right to left) succession (Fig. 11). At the onset of shortening, syn-contractional extension occurs in the foreland, as in Model 2 (see Supplemental Material for Model 3). Pre-contractional model salt diapirs extruded salt onto the model surface (Fig. 11a). T1 develops in the hinterland proximal to the backstop with an arcuate thrust front. Unlike Model 2 where two thrusts (T1 and T2) develop in the hinterland proximal to the backstop, only one thrust (T1) develops in this region (Figs. 8a, 11a). Only the first two syn-contractional layers are deposited in the hanging wall of T1. As T1 continues to propagate forward, a graben occurs at the crest of the thrust-related anticline. Time-lapse videos reveal that salt wall squeezing occurs at about 6 cm (~25% of shortening) across the entire model (Fig. 11a). The squeezed salt walls nucleate backthrusts (BT2) and thrusts (T3) at around 12 cm (~50% of shortening), and the syn-contractional sedimentary wedge continues to prograde towards the foreland (Fig. 11b). Downbuilding occurs across the accommodation zone in response to continued extrusion of model salt from isolated salt plugs (Fig. 11b). By 18 cm of shortening, the individual thrusts nucleated along the salt walls have merged into a single thrust front with small backthrusts and pop-ups developing in the hanging wall (Fig. 11c). The height of the thrust front of Thrust 3 is great enough to restrict deposition of syn-contractional sand to its footwall. Continued degradation of the thrust front of Thrust 1 and syn-contractional extension on its hanging wall led to model salt piercing in its thrust front near Backthrust 2 (Fig. 11c). This led to model salt extrusion that, with further shortening, extends laterally along the Thrust 1 thrust front, breaking through more of the front toward the top of the model (Fig. 11d). The foreland pop-up structure, bounded by Thrust 4 and Backthrust 4, develops at around 20 cm of shortening and creates enough elevation to restrict deposition of the final syn-contractional sedimentary layer in the footwall of Thrust 4, along the top of the model, and in the top half of the footwall of Backthrust 4 (Fig. 11d). The asymmetry in the deposition of this final layer of sand is attributed to the large volume of extruded model salt coming from salt walls along the model edge. The thrust chronology chart displays a more regular pattern of thrust initiation timing with a roughly linear trend (Fig. 11e).

The top basement structure map reveals four structural steps that have decreasing average elevations from right to left (Fig. 12a). The third basement thrust sheet level incorporates Master Fault 1 and Master Fault 2. Two individual thrust sheets develop

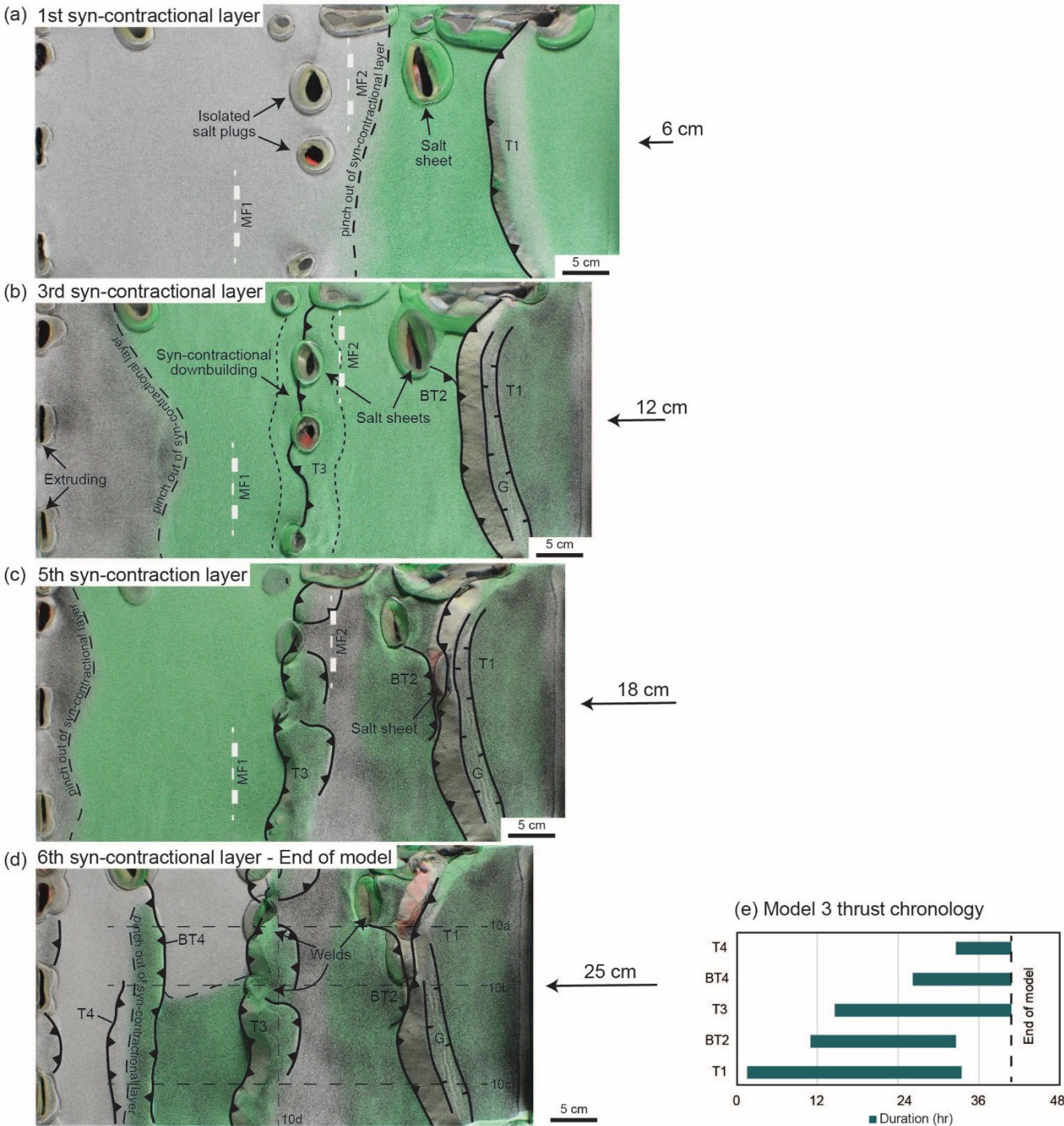

Figure 11. (a-d) Top-view images illustrating the contractional kinematic evolution of Model 3 (shortening with sedimentation), including the timing of thrust initiation, model salt extrusion from salt sheets, contractional welding of salt sheets, and the distribution of syn-contractional sedimentation. (e) Graph illustrating the thrust chronology of the major thrusts and backthrusts. Abbreviations: BT – backthrust; G – graben; MF – Master Fault; P – pop-up structure; T – thrust.

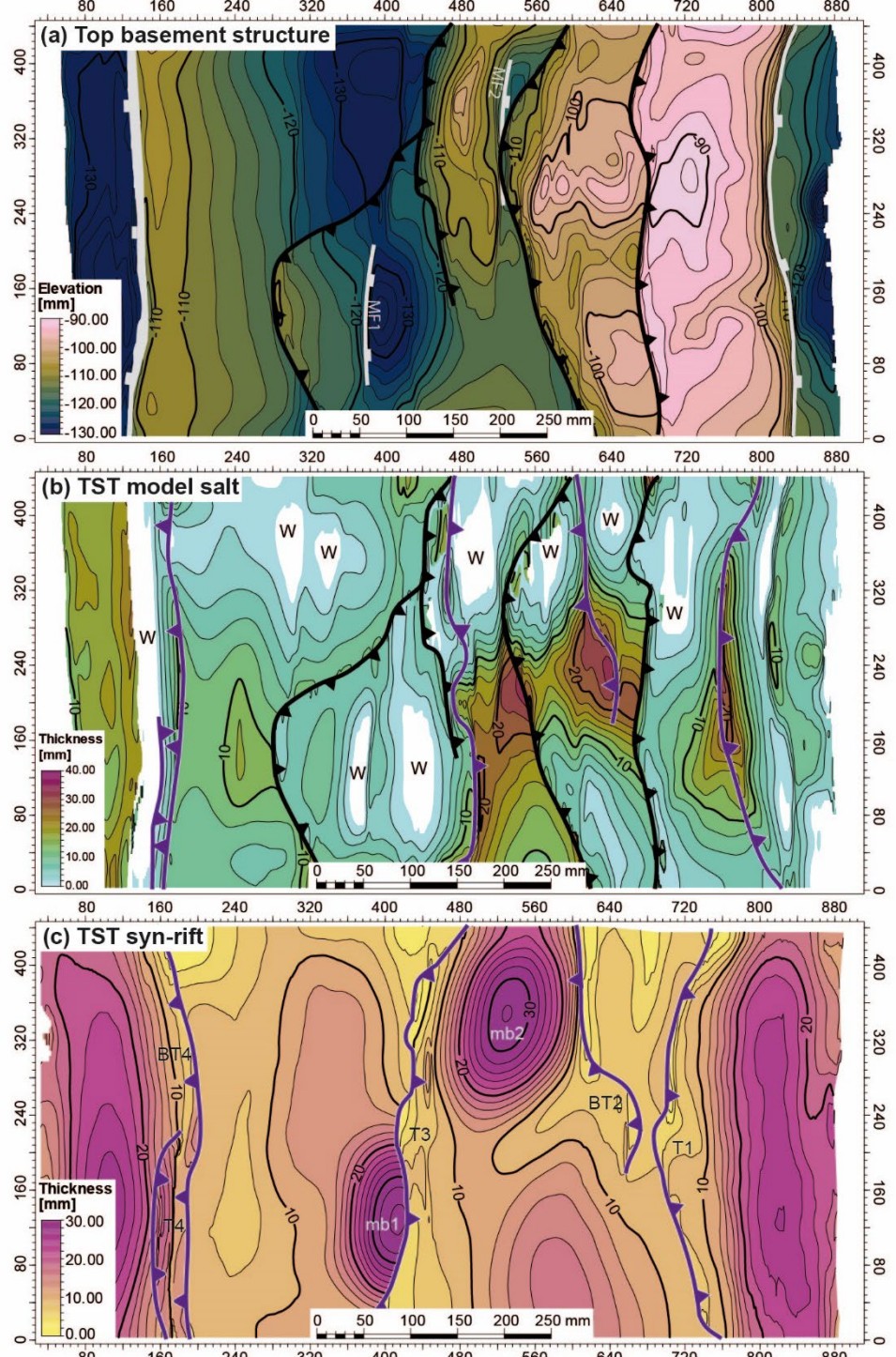

Figure 12. Structure and true stratigraphic thickness (TST) maps of key horizons from the voxel results of Model 3 (shortening with sedimentation): (a) Top basement structure map of Model 2, (b) TST syn-rift map, and (c) TST autochthonous salt map. Basement

**thrusts are in black and cover thrusts are in dark magenta. Abbreviations: BT – backthrust; MF – Master Fault; mb – minibasin; T – thrust; W – weld.**

on the foreland side of each main normal fault and associated half graben; a lateral ramp in the thrust sheet ahead of Master Fault 1 links the thrust sheet with the thrust ahead of Master Fault 2.

The map showing the distribution of salt thickness highlights the relationship between the model salt and the thrust systems (Fig. 12b). As in Model 2, the true stratigraphic thickness salt map reveals that the number of welds increases two-fold during shortening. The pre-contractional primary welds along minibasin 1 and minibasin 2 are more extensive and the welded along the foreland and hinterland sides of both minibasins (Fig. 12b). New primary welds are present along the top half of the model and correspond with the area of significant syn-contractional model salt extrusion (Figs. 11, 12b) and with the areas of highest elevation along the basement thrust sheets in the hinterland (Figs. 12a, 12b). Like Model 2, the basement thrusts leave clear imprints on the final model salt distribution and the thrust fronts can easily be interpreted across the thickness map. The cover thrust system leaves a less obvious influence on the final model salt distribution. Salt anticlines and salt walls appear as steep-sided structures in the model salt true stratigraphic thickness map, below the cover thrusts (Fig. 12b). The largest amount of preserved model salt across the basement accommodation zone occurs below the hanging wall of Thrust 3 (Figs. 10a-b, 12b).

The true stratigraphic thickness map of the syn-rift succession reveals the location of the syn-rift depocenters and their location in relation to the cover thrust system (Fig. 12c). Unlike Model 2, the depocenters do not consistently remain in the footwalls of the thrust sheets. The left-hand (foreland) marginal minibasin forms the hanging wall of Backthrust 4. However, no thrust develops ahead (forelandward) of minibasin 1 and, therefore, minibasin 1 remains in the footwall of Thrust 3 at the end of the model. As in Model 2, the thinnest part of the syn-rift succession is adjacent to the interpreted thrust fronts of each of the faults in the cover thrust system (Fig. 12c).

## 4. Discussion

In the following section, we analyze the influence of salt in the formation and inversion of laterally segmented salt-bearing rift basins. Our analogue models are then compared to natural examples of salt-bearing rift basins inverted to different degrees, namely the northern Lusitanian Basin of Portugal and the Southern Pyrenees of Spain.

### 4.1 Extensional deformation and syn-rift sedimentation in salt-bearing segmented rifts

In our models, the syn-rift basin geometry and related sediment distribution are strongly controlled by the position of the underlying pre-salt basement faults, the amount and rate of slip attained by those faults, the original distribution and thickness of model salt, as well as the thickness and mechanical properties of the analogue modeling materials (i.e., the sand and the polymer, see Table 2) (e.g., Withjack et al., 1990, Withjack and Callaway, 2000; Ferrer et al., 2014, 2016, 2022; Tavani and Granado, 2015; Roma et al., 2018a; Granado et al. 2021). In our models, syn-rift faults nucleated over velocity discontinuities imposed by the experimental set-up: at the lateral basal polymer pinch-outs, at the transition points between the three basal materials, and at the edge of the silicone seeds that controlled the location of the major faults (Fig. 2). The triangular section of the silicone seeds constrained the nucleation of listric master faults (i.e., Master Fault 1 and Master Fault 2). Gentle differences are found regarding fault slip and curvature of the major faults, likely related to the proximity of the model salt seeds with respect to the velocity discontinuities. Smaller planar extensional faults develop along the other velocity discontinuities. The metal plate contact acts as a more effective discontinuity, creating extensional faults with greater offset than those faults that nucleate along the rubber- and mylar-sheet contacts. The relatively greater offset created by the thickness of the metal plate (2 mm) compared to the thinness of the mylar sheet (<1 mm) may thus relate to differences in fault nucleation effectiveness. Additionally, the velocity discontinuity along the metal plate boundary is proximal to the fault tip of Master

Fault 1 and the underlying silicone seed (Fig. 6c). It is likely that the stresses are not uniform along the silicone seeds, as they develop half

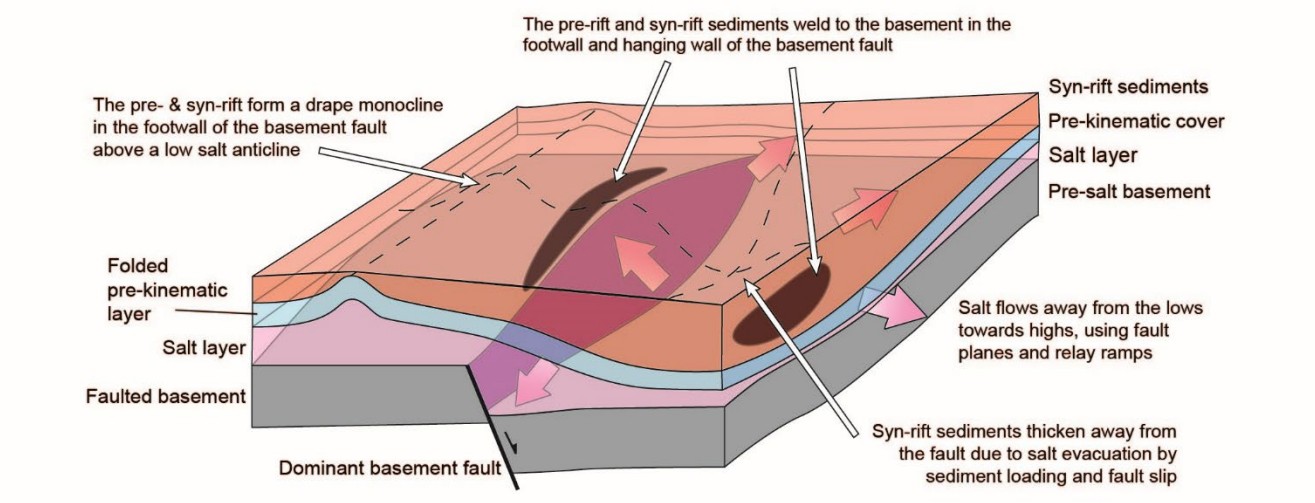

**Figure 13. Block diagram illustrating the structural style of decoupled extensional rift half-graben with thick early syn-rift salt and a thin pre-kinematic layer bed showing the migration and inflation of salt away from the half-graben and the development of a synclinal minibasin within the basement half-graben. Modified and updated from Richardson et al (2005).**

grabens with right-dipping master faults on the lefthand side and minor faults on the righthand side, in section view (Figs. 4a,c), and this could also influence the effectiveness of the metal plate velocity discontinuity as it is on the lefthand side of the model set up and proximal to Silicone Seed 1 (Fig. 2). In our models, the large syn-rift extensional faults grew by the lateral propagation of their tips, in a rather similar fashion to natural systems (e.g., Peacock and Sanderson, 1991; Cartwright et al., 1995), whereas across the accommodation zone in between, a more diffuse fault array develops and forms a horst in cross section (Fig. 4b). However, this horst structure forms a saddle in map view with a lower elevation than the adjacent basement along strike (Fig. 6a). The results of our modeling program highlight two key points: the relative timing of fault activity and the intervening phases of syn-rift sedimentation have a strong impact on the geometry of syn-rift basins, and that the thick (i.e., 1cm minimum) layer of syn-rift salt deposited after the first stage of rifting favored the complete decoupling of extensional deformation between the pre- and the post-salt extensional systems. As a result, two very different extensional deformation styles developed. Similar results have been obtained in analogue models by Dooley et al. (2003), Vendeville et al. (1995), Withjack et al. (1990), and Withjack & Callaway (2000). The model salt and cover sequence accommodate basement extension via stretching and thinning. The interplay between basement extension and syn-rift sedimentary loading initiate salt migration from the basement lows towards the margins of the sedimentary depocenters, as observed as well in previous works (e.g., Ferrer et al., 2014; Granado et al. 2021; Hudec & Jackson, 2007; Kehle, 1988; Koyi et al., 1993; Roma et al., 2018a; Dooley and Hudec, 2020). This creates a positive feedback loop that accelerates the migration of salt from the basement lows as the subsiding minibasins thicken and sink deeper (e.g., Santolaria et al. 2021a). However, the dimensions and location of the depocenters for minibasin 1 and minibasin 2 do not reflect the underlying basement geometries. In cross- and strike-sections, the syn-rift minibasins have symmetrical synclinal geometries that hamper the interpretation of the location and polarity of the underlying basement faults (Fig. 4), and thus contradict the assumption that the geometry of sedimentary fill in the hanging wall of a fault is directly related to the shape of the normal fault (i.e., White et al. 1986). The margins of the minibasins extend beyond the underlying downthrown basement blocks, masking the locations of the normal faults (Figs. 4a, c; 6b). Without prior knowledge of the basement tectonic history, it would be very difficult to link the kinematic evolution of the cover depocenters to the underlying basement extensional system. Yet the depocenters of the syn-rift minibasins develop adjacent to the underlying basement faults and their locations still correlate with the displacement maxima of the hanging wall block (Figs.

4a, c; compare Fig.1 and Fig.13). The thickness map of the syn-rift succession highlights that the areas of thinner stratigraphy correspond with areas of underlying model salt inflation in the form of pillows, anticlines, and salt walls across the

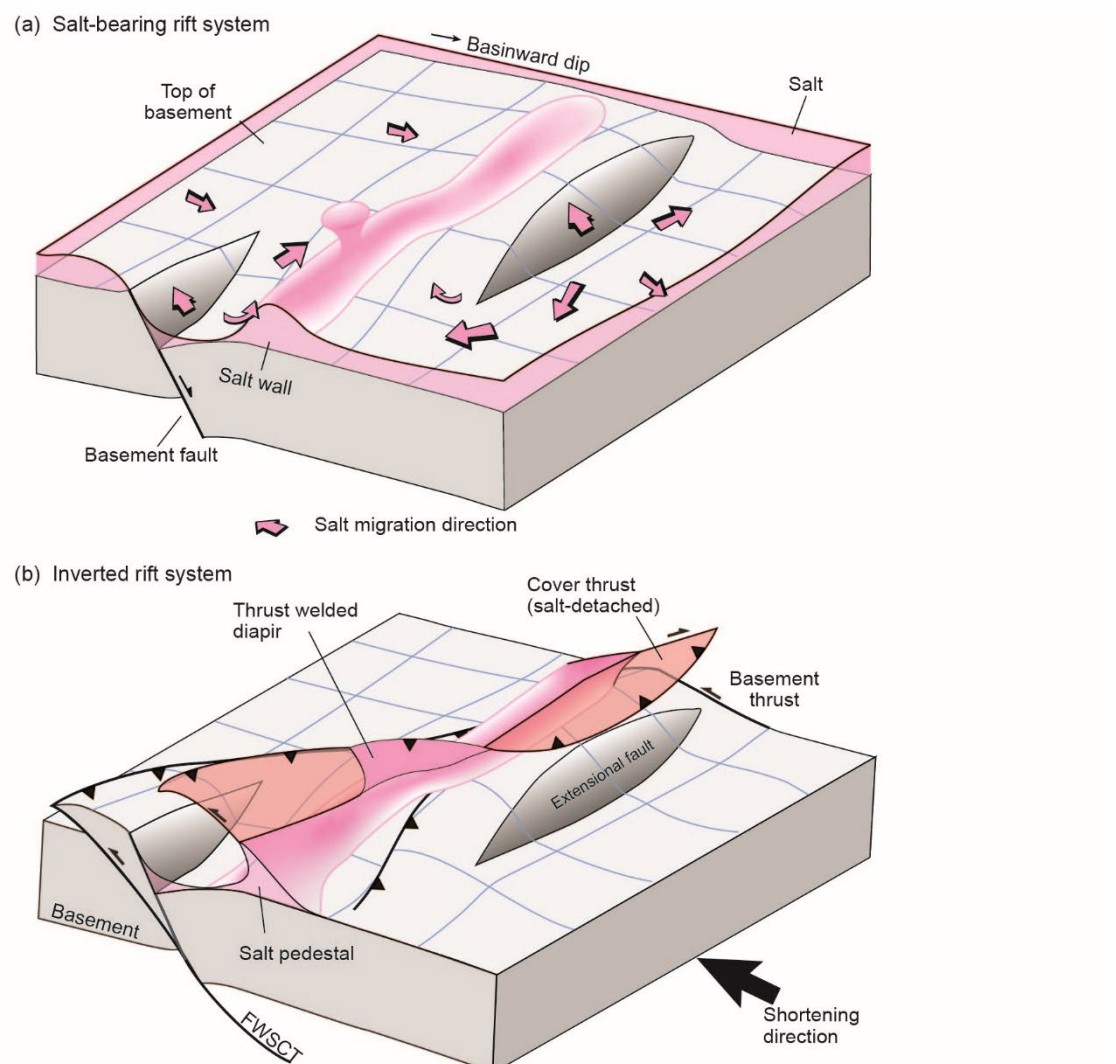

 **Figure 14. 3D schematic block diagrams illustrating the relationship between basement fault extension and the migration of salt caused by extension and downbuilding of cover strata (a), and how the salt interacts with the underlying basement thrust system and controls the location of thrusts through the cover sequence during subsequent shortening (b).**

accommodation zone, while areas of non-deposition/erosion correspond with locations where inflated salt has pierced the cover sequence and spread over the model surface (Fig. 5). These results are important for the interpretation of seismic data, and the location of wells in pre-salt targets. In our models, areas of salt inflation are found over the footwall blocks away from the underlying basement faults (e.g., Koyi et al., 1993; Withjack & Callaway, 2000; Dooley and Hudec, 2020; Granado et al., 2021), but also along the accommodation zone as an elongate, slightly sinuous salt anticline crossing from the hanging wall of Master Fault 1 to the footwall of Master Fault 2. In this salt ridge across the accommodation zone, the thickest salt accumulation corresponds to the salt wall over the hanging wall of Master Fault 1. Remnant salt is trapped against the fault planes of the master faults and the overlying hanging wall cover succession (Figs. 4a-c and 13).

Based on the results of our experimental program, we have updated the 3D block diagram from Richardson et al. (2005) to summarize the key salt structures, cover depocenters, and other features related to the interplay between basement extension, salt migration, and syn-rift sedimentation (Figs. 1 and 13). The model proposed by Richardson et al. (2005) includes a comparatively thicker pre-kinematic layer than that used in our analogue models. A thinner pre-kinematic layer facilitates the development of shorter wavelength and higher amplitude forced fold across the basement fault, as shown in earlier works (e.g., Withjack, and Callaway, 2000). One relevant implication conerns the lateral boundaries of hanging wall minibasins, and the distribution of the syn-rift sedimentary infill: minibasins with high-amplitude and short wavelength force-folded pre-kinematic layers would favor the vertical aggradation of downbuilt syn-rift sediments, while longer wavelength and shorter amplitude force-folded layers should favor the lateral expansion of syn-rift sediments. This model also shows that the syn-rift depocenter is shifted away from the basement fault. The model by Richardson et al. (2005) shows a forced fold with four-way dip closures on the footwall of isolated basement faults; syn-rift layers onlap the periclinal termination of these footwall anticlines, resulting in a syn-rift sediment pinch-out with a concave to the fault arcuate geometry (Fig. 1). Conversely, our models show a continuous salt ridge stretching from the hanging wall of Master Fault 1 to the footwall of Master Fault 2 across the accommodation zone (Fig. 14a), precluding the formation of an isolated four-way dip closure anticline on the footwall of Master Fault 2. This results in different syn-rift pinch-out geometries, which are rather elliptical in plan-view (Fig. 6b). Minibasin 1, however, fits better with the model proposed by Richardson et al. (2005), since its footwall has not interacted with any other fault (due to the model set up limitations). A diapir formed along accommodation zone by a combination of stretching of the overburden and erosional piercement atop inflated salt. This is evidenced by the unconformity between pre-kinematic and syn-kinematic layers at both flanks of the structure (Fig. 4c). A salt sheet extruding towards subsiding minibasin 1 formed during the end of rifting (Fig. 5c) and grew passively during the deposition of the post-rift sand pack (Fig. 4c). Extrusion of model salt was probably enhanced by the impinging of the overburden above the footwall of Master Fault 1 forming a primary weld, as Stewart (2017) and Ferrer et al. (2022) have pointed out in previous works. The occurrence of these large salt walls and local salt extrusion across accommodation zones have also important implications for the subsequent shortening phases, and in particular, for the orientation of contractional structures in respect to the regional direction of shortening.

## 4.2 Influence of salt and syn-contractional sedimentation on contractional structural styles and kinematics

In our models 2 and 3, thick syn-rift model salt and associated structures, and the presence of syn-contractional sedimentation played key roles in the structural styles and kinematics of the contractional fault systems. In both models, thick syn-rift model salt controls the structural style of the cover thrust system and accommodation of contractional deformation ahead of the moving backstop wall. Syn-contractional sedimentation is added as an additional controlling factor in Model 3. In this way, Model 2 can be used as a control model for contractional deformation to differentiate the effects of the model salt décollement on structural style and thrust kinematics from those imposed by syn-contractional sedimentation.

Both models have a generally foreland propagating thrust style in the basement and cover systems. As during extension, the model salt decoupled the basement and cover thrust systems for the duration of contraction, except in locations of primary welding. In those positions, however, the thrust systems did not link via faulting. Comparison of true stratigraphic thickness model salt maps for models 1-3 (Figs. 6b, 9b, 12b) suggest that additional primary welds developed during shortening, attributed to basement thrust uplift and propagation, as well as syn-contraction model salt extrusion. Reactivation of the main extensional faults during shortening has been achieved by the nucleation of the basement thrusts in the silicone seeds. The degree of inversion is higher in Master Fault 2 than in Master Fault 1. A likely explanation is the distance between the master faults and the deformation front (i.e., the backstop): Master Fault 2 is closer to the backstop and accommodates much more shortening than Master Fault 1 (e.g., Jara et al., 2015). In the section including Master Fault 1, basement deformation is more distributed than in the section through Master Fault 2. This style of deformation propagation was not seen in Dooley and Hudec (2020) because their models were compressed from both sides, using both moving walls, resulting in a culmination of contractional deformation in the middle of the model. The other minor preexisting extensional faults were not reactivated as

thrusts, as is common in analogue models built using sand layers (e.g., Eisenstadt and Simms, 2005; Ferrer et al., 2014; Granado et al. 2017). Instead, low-angle thrusts detached on the basal silicone layer, crosscutting the preexisting extensional fault system (Figs. 7, 10), similar to the inversion style in Dooley and Hudec (2020).

As proven by previous studies, areas of inflated salt (e.g., salt anticlines, walls, and plugs) absorb and nucleate contractional deformation (e.g., Vendeville et al., 1993; Vendeville and Nilsen, 1995; Letouzey et al., 1995; Brun and Fort, 2004; Rowan 665 and Vendeville, 2006; Hudec and Jackson, 2007; Dooley et al., 2009; Duffy et al., 2018; Roma et al., 2018a; Dooley and Hudec, 2020; Granado et al., 2021; Santolaria et al., 2021a, b; 2022). In our models, this process began in the early stages of shortening, as evidenced by narrowing of salt diapirs (Figs. 8, 11, and Supplemental Material). Our results indicate that the geometry of salt diapirs and salt structures dictates the nucleation of thrusts, and strongly influences the subsequent thrust kinematics. Diapirs preferentially developed in the accommodation zone between the two major extensional faults and along 670 the salt ridges. They record the onset of shortening in the foreland away from the thrust front and determined the linkage of the thrusts reactivating the major extensional faults as contractional deformation progressed forward (Figs. 8 and 11). In Model 2 with no syn-contractional sedimentation the squeezed diapir of the accommodation zone linked with the backthrust related to the reactivation of Master Fault 2 and the thrust squeezing the salt structure in the hanging wall of the reactivated Master Fault 1. As a result, the change of vergence and the oblique structures connecting with the welded diapir revealed the existence 675 of a former accommodation zone in the basement faults, regardless of the decoupling between the basement and the post-salt succession. In Model 3, syn-contractional sedimentation in the footwall of the reactivated Master Fault 2 prevented the development of the backthrust in front of minibasin 2, resulting in a more linear structure and masking the existence of the accommodation zone (Figs. 10a and 11). This linearity has been enhanced by the along strike coincidence of the salt anticline above the rollover anticline in the hanging wall of Master Fault 1 and the salt anticline in the footwall of Master Fault 2, 680 regardless the saddle between both anticlines coinciding with the accommodation zone (Figs. 4, 5 and 11).

Random thrust vergence (i.e., fore thrusts *vs.* back thrusts) is one characteristic behavior of thrust wedges formed on non-frictional décollements (e.g., Davis and Engelder, 1985; Costa and Vendeville, 2002). However, several factors in our models may have also favored the development of oppositely verging thrusts, particularly for those structures showing a sudden change in vergence along strike (i.e., Model 2), including: the location of primary welds relative to the squeezed salt wall, the 685 distribution of the cover sequence both across and along the salt wall, and the width of the salt wall. The thickness of the original model salt, and consequently, the thickness of the cover sequence, seem to be one controlling factor as the thick minibasins tend to remain on a footwall position (Fig. 7). This is contradictory to what is expected to happen upon positive basin inversion, in which the former extensional basin is uplifted and thrust above the thinner succession in the former extensional footwall (e.g., Cooper et al., 1989; Williams et al., 1989). Isolated sub-circular plugs nucleate foreland-propagating 690 thrusts that tend to link up if other thrusts initiate along strike, in the same structural position (Figs. 11b-d). It has been proven in many experimental programs that the geometry and the position of emergent thrusts is strongly influenced by erosion and sedimentation processes (e.g., Storti and McClay, 1995; Mugnier et al. 1997; Graveleau et al. 2012). Conversely, recent works propose that the location and distribution of the early salt structures and related syn-rift minibasins, determine the location, wavelength, and vergence of thrusts in the cover system, regardless of the presence of syn-contractional sedimentation (e.g., 695 Roma et al., 2018a; Dooley and Hudec, 2020; Santolaria et al. 2021a, b). In our models, however, the difference in the pattern of thrust vergence without (Model 2) and with syn-contractional sedimentation (Model 3) is striking.

It is also remarkable that both models show significant differences in the geometries of basement thrusting, and the associated distribution of structural relief, and thrust sequence. These differences evidence the role imposed by syn-contractional sedimentation in controlling the large-scale kinematics of thrust wedges (i.e., cover and basement; e.g, Storti and McClay, 700 1995; Fillon et al., 2013; McClay, 2015; Lacombe et al., 2019; Pla et al., 2019), since the presence of early salt structures and syn-rift minibasins is broadly equal on both models. In our models, the location and geometry of the syn-rift salt structures reflect the inherited extensional basement fault array, which due to the model set-up run broadly perpendicular to the direction of extension and shortening. However, the observed relationships have been transported by thrusting: this is important to note because it could help reconstructing the pre-shortening configuration of a salt-bearing rifted margin that has been detached its

original position, or in areas where the sub-salt configuration is obscured. Note as well that in such a case, the geometry of the syn-rift minibasin cannot not be used to infer the polarity of the basement fault array (i.e., whether left- or right-dipping, in model coordinates). These insights can also be applied to other salt-bearing systems, such as the Jura Mountains (e.g., Rime et al., 2019; Schori et al., 2021) and SW French Alps (e.g., Soto et al., 2017; Célini et al., 2020)

Syn-contractional sedimentation was added to Model 3 as a prograding sedimentary wedge that onlapped the underlying cover sequence. This sedimentary wedge triggered syn-contractional downbuilding, enhanced model salt extrusion due to salt evacuation, and altered the rate of transfer of contractional deformation from what was seen in Model 2. The syn-contractional sedimentary wedge renewed downbuilding via differential loading. The syn-contractional were folded into footwall synclines beneath emergent thrust fronts, forcing salt evacuation from below the footwalls, as well as enhancing salt extrusion from preexisting salt structures by combining syn-contractional extrusion with downbuilding. Local areas of higher model salt extrusion are reflected by the uneven thicknesses and distribution of the syn-contractional sedimentation across Model 3 (compare Figs. 7a and 7c; Fig. 11), and the true stratigraphic thickness map of the model salt (Fig. 12c).

The missing model salt in the hanging wall of the cover thrusts proximal to the thrust fronts in Model 2 could also be related to the difference in the rate of transference of contractional deformation into the foreland between models 2 and 3. There is a difference in the thrust sequencing in cover thrust systems between models 2 and 3. There is a delay in the propagation of thrusts towards the foreland with syn-contractional deformation. In addition, more backthrusts developed with no syn-contractional sedimentation (Figs.7, 8, 15a; see Supplemental Material). The out of sequence thrust system in Model 2 indicates that deformation was transferred rapidly across the model, utilizing any and all weak points to nucleate thrusts (Fig. 8e). None of these structural features are seen in Model 3. In fact, the timing of thrust nucleation in the cover system has a relatively linear relationship with time (Fig. 11e). We attribute this change in thrust nucleation timing with the addition of syn-contractional sediment at regular intervals. The syn-contractional sedimentary wedge changed the taper angle of the thrust wedge, focusing deformation in the hinterland until critical taper was once again attained, allowing contractional deformation to propagate farther into the foreland.

Based on the results of the shortened models, we propose a model illustrated by 3D block diagram showing the inversion of a salt-bearing segmented rift (Fig. 14b). During shortening, the salt inflated areas localize shortening via squeezing and as décollements for thrusts in the cover (Fig. 14b). Extrusive salt diapirs localize thrust fronts while elongated salt walls nucleate thrusts with opposing vergence, regardless of the presence of syn-contractional sedimentation. Inversion of the basement system results in the preferential development of footwall shortcuts across preexisting extensional faults. Individual thrusts may join by segment linkage to form oblique ramps across extensional accommodation zones.

### 4.3 Insights for natural systems

The results of analogue models of an inverted segmented rift system with early syn-rift salt are compared with natural examples from the northern Lusitanian Basin in offshore Portugal and the South-Central Pyrenean fold and thrust belt in Spain (Fig. 16). The three natural case studies show increasing amounts of inversion and shortening, from very little in the northern Lusitanian basin, to full basin inversion and incorporation in the thrust system in the Southern Pyrenees case.

### 4.3.1 Northern Lusitanian Basin, offshore Portugal

The northern Lusitanian Basin offshore Portugal records several multirift events, from Late Triassic to Early Cretaceous, related to the opening of the North Atlantic (e.g., Alves et al., 2002; Rasmussen et al., 1998; Stapel et al., 1996). The northern Lusitanian basin is also mildly inverted as a result of the Alpine orogeny, with a relatively minor Eocene compression event followed by a more pronounced period of Miocene deformation, characterized by folding, minor faulting and erosion (e.g., Pinheiro et al., 1996; Ribeiro et al. 1990). According to these authors, the Dagorda salt of the northern Lusitanian basin was deposited during Triassic rifting, pointing out the occurrence of salt migration during the Triassic, Jurassic and Early

Cretaceous rifting events. In the salt-prone regions of the basin, broad minibasins developed above the hanging wall blocks of pre-salt basement faults, forcing salt to inflate on the footwall blocks of basement faults (see Alves et al., 2002; Fig. 15a), probably aided by differential sediment loading. Thus, the northern Lusitanian basin is comparable to our analogue models since the origin of salt structures can be related to the interaction of pre-salt basement fault kinematics and syn-rift
sedimentation in the presence of thick syn-rift salt.

In the interpreted seismic profile (Fig. 15a), the cover sequence is characterized by a broad salt-withdrawal minibasin affected by several salt-detached folds above an uneven basement. Salt-detached cover folds are located at the edges of the main syn-rift depocenter, while a larger anticline is observed in the middle of the seismic profile. The syn-rift sequence is characterized by a series of reflectors that apparently downlap on the pre-kinematic series towards the SW of the profile, while the same
reflectors onlap on the pre-kinematic units towards the NE end of the seismic line. These reflector terminations are broadly similar to those produced in our sandbox models and are indicative of extension by underlying faults while sediment loads and

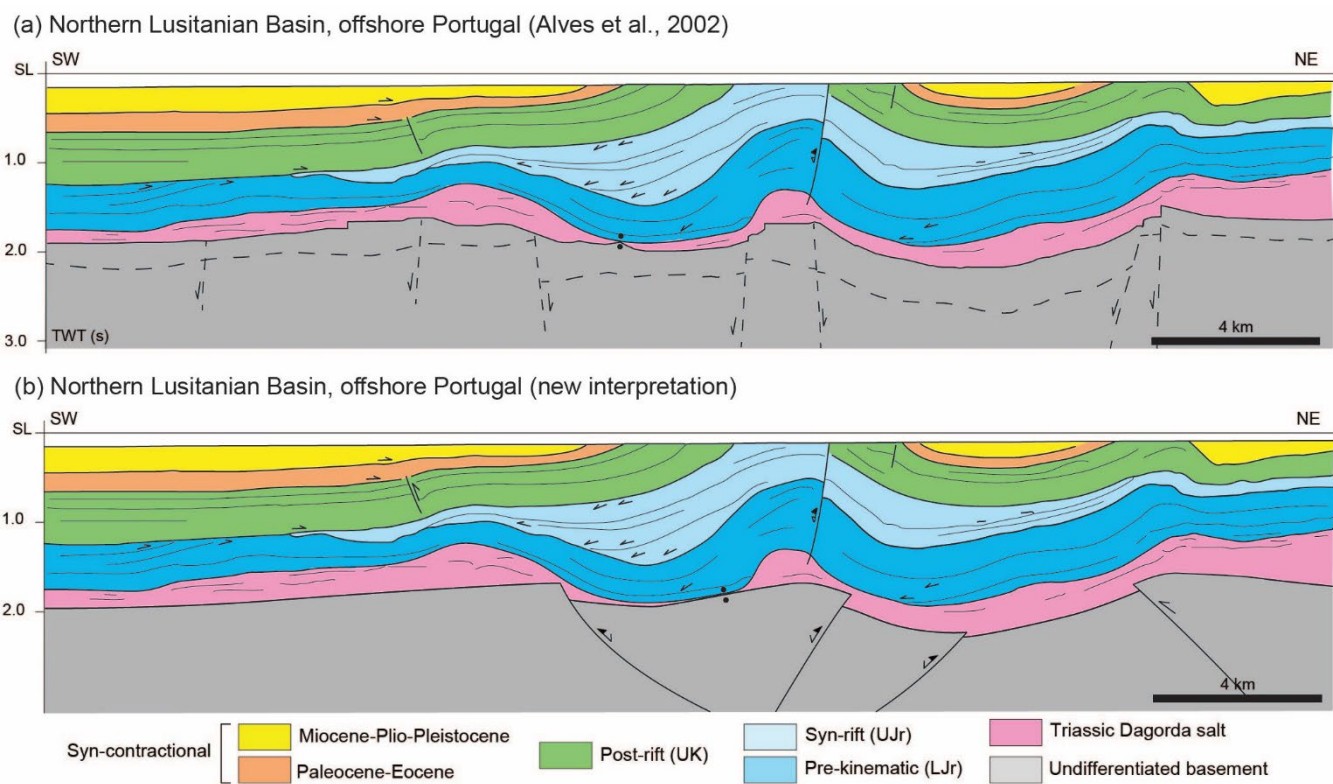

Figure 15. Line drawings of seismic sections showing variable degrees of inversion of salt-detached extensional systems. (a) Interpreted seismic line S84-23 of the northern Lusitanian Basin (offshore Portugal) (modified from Alves et al., 2002); (b) New
interpretation of the basement of seismic line S84-23. Abbreviations: Tr: Triassic, Jr: Jurassic; K: Cretaceous; L: Lower; U: Upper.

evacuates underlying salt. The apparent downlap most probably results from the folding and inversion of previous onlap geometries, as also shown by previous authors in other salt-bearing rift basins (e.g., Tavani and Granado, 2015) and numerical models (e.g., Granado et al. 2021).

While there is good enough imaging to define the base salt in this section of the northern Lusitanian Basin, the basement
configuration below the salt is not well imaged, which leaves the structural style of the basement open to question. Differing interpretations have been provided to explain the observed geometries of the inverted post-salt extensional basins. For instance,

Alves et al. (2002) originally interpreted these syn-rift basins as salt withdrawal minibasins mostly inverted by thin-skinned shortening, with the evaporites acting as a regionally decoupling horizon. The implications from such interpretation, as shown by our models, is that if pure thin-skinned shortening has taken place, the pre-salt basement faults responsible for crustal thinning and syn-rift accommodation space may not be directly below the salt structures and related cover folds. The salt structures and related depocenters may have been transported along the decoupling horizon, and thus laterally shifted from the underlying basement fault array. Alternatively, Roma et al. (2018b) interpreted the individual syn-rift basin as a salt-detached ramp syncline basin, whose tectonic inversion was accomplished by the reactivation of an underlying basement fault with a ramp-flat-ramp geometry. The sections of our models (Figs. 7, 10) may provide a glimpse towards a more evolved and inverted equivalent to the northern Lusitanian basin. However, certain improvements can be made in the interpretation of the seismic profile based on our models: the presence of basal primary welds, not originally interpreted by previous authors (i.e., Alves et al., 2002, their Fig. 10; our Fig. 15a), for instance. The uplift and basinward (SW) rotation of the syn-rift depocenter in the central portion of the profile could be attributed to salt squeezing, mild inversion of a normal fault in the cover sequence (Alves et al., 2002), the mild inversion of preexisting normal faults and, most likely some combination of the two as can be seen in our models. Taking these concepts from the results of our models, the pre-salt basement in the seismic profile could be reinterpreted to show a primary welded syn-rift basin above reactivated basement faults (Fig. 15b).

### 4.3.2 Isàbena area, South-Central Pyrenees, Spain

The Pyrenean orogen formed during the Late Cretaceous to Miocene collision of the Iberian and European plates (Muñoz et al., 2018 and sources therein). The South-Central Pyrenean fold-and-thrust belt is part of the southern thrust system of the doubly verging Pyrenean orogen. It is characterized by the inversion of the Late Jurassic-Early Cretaceous salt-bearing segmented rift system developed along the North Iberian Margin (Tugend et al., 2014; Tavani et al. 2018). The fold-and-thrust belt is characterized by a series of thrust sheets that involved a Mesozoic succession decoupled it from its autochthonous basement along the Triassic evaporites (Fig. 16). Available seismic lines and tied exploration wells allowed for a constrained interpretation of key structural features of the fold-and-thrust belt beyond extensive surface data (Muñoz, 1992; Muñoz et al., 2018). Unlike the previous two examples, the South-Central Pyrenees are an example of a fold-and-thrust belt that has undergone more inversion and shortening than our analogue models as the supra-salt rift system has been completely detached and transported from its original position and can no longer be correlated to the underlying basement structures responsible for crustal thinning.

Several examples of inverted syn-rift basins have been interpreted in the South-Central Pyrenees, including the Organyà, Aulet, Las Aras, and San Juan basins (Muñoz, 1992; Garcia-Senz, 2002). We have selected the Isàbena area to illustrate key similarities in the geometries of structures within the Pyrenean example and our models (Fig. 16). Two syn-rift basins (i.e., Las Aras and San Juan basins) have been inverted and transported in a salt-detached thrust system while apparently maintaining their original segmented configuration (Fig. 16a). The zone between the two syn-rift basins is defined by the development of a large post-rift basin (the Cotiella basin) and a series of oblique salt-detached anticlines, flanked by growth synclines filled by Upper Cretaceous syn-contractional turbidites (e.g., Garcia-Senz, 2002; Ramos et al., 2020). Very little of the San Juan basin is preserved as a result of uplift and erosion, but some salt is preserved in the contact with the Cotiella basin to the SW, which can be interpreted as a squeezed diapir that developed in the footwall of a former extensional fault (Fig. 16b). The Las Aras basin is clearly inverted and characterized by an anticline along the basin's southwestern margin. To the north, Albian syn-rift strata are onlapping previously eroded Jurassic dolostones (Garcia-Senz, 2002) thus suggesting that the Jurassic strata were forced folded upon rifting, favored by the decoupling of the Triassic evaporites. To the NE, the Las Aras basin is secondary welded to the Alins syncline, so it is likely that interaction with a growing salt diapir along the northern margin of the basin controlled the deposition of the syn-rift succession.

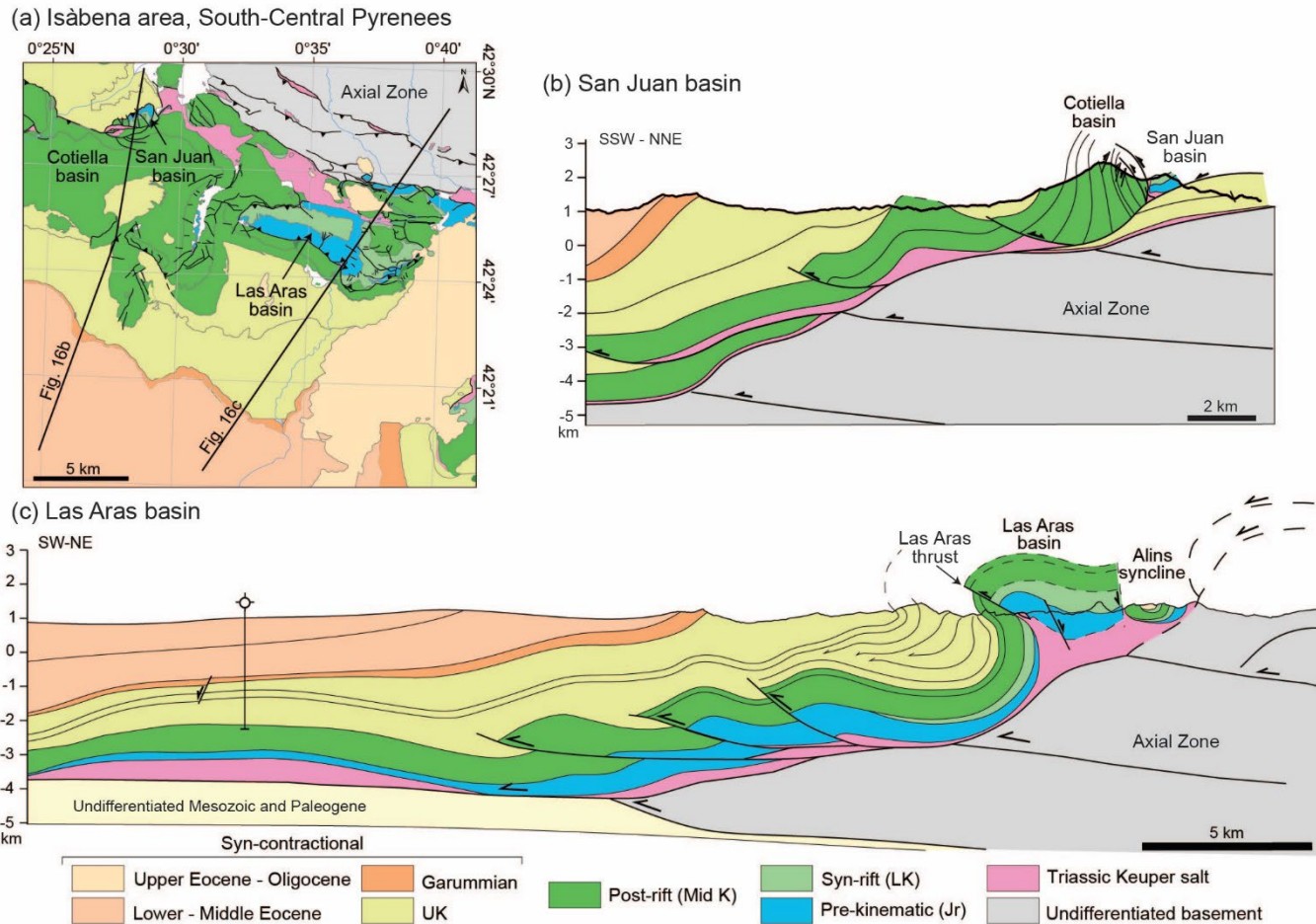

Figure 16. (a) Map showing the location of cross-sections through the San Juan and Las Aras basins, South-Central Pyrenees (compiled from Robador and Zamorano, 1999; García-Senz and Ramírez, 1997). (b) Cross-section through the San Juan basin. (c) Transect across the Las Aras basin (modified from Muñoz et al., 2018). Abbreviations: Jr: Jurassic; K: Cretaceous; L: Lower; U: Upper.

Salt tectonics continued to influence the system into the post-rift. The post-rift succession is fairly uniform across the section through Las Aras (Fig. 16c). However, in the section through the San Juan basin and to the west (Figs. 16a, b), the Cotiella basin developed as post-rift gravity driven extensional basin above the Triassic salt in the footwall of the San Juan basement fault associated with the San Juan syn-rift basin (López-Mir et al., 2015). As a result of contractional deformation, the cover system (Jurassic pre-kinematic through Mid Cretaceous post-rift) containing the syn-rift basins has been detached on the Triassic saltand transported some tens of kilometers from its original position in the Pyrenean rift margin, without the basement rift system (Séguret, 1972; Muñoz, 1992; Puigdefàbregas et al., 1992; Teixell, 1996), so it is difficult to determine the degree of coupling (if any) between the cover syn-rift basins and underlying basement extensional fault array prior to detachment as there is no evidence that the basement extensional fault system has been transported with the cover fold and thrust belt. Based on comparison with our analogue models, we can infer that the basement master faults and associated half-grabens (or grabens) were once located beneath the Las Aras and San Juan basins, providing the accommodation space required to develop the Lower Cretaceous syn-rift basins in their hanging walls. And assuming that the cover system has been transported in such a way to maintain their original relational positions, the two syn-rift basins were separated by a wide accommodation zone.

Today, the former extensional accommodation zone is characterized by the inverted Cotiella basin and a tight, obliquely oriented salt-cored anticline of pre-rift and post-rift units that extends to the south of the Las Aras basin, flanked by the Upper Cretaceous syn-contractional sediments.

The rate and amount of syn-contractional sedimentation is much higher in this Pyrenean example than in our models, as can be seen by the thick syn-contractional succession burying the underlying thrusts under over 1 km of sediment (García-Senz, 2002; Muñoz et al., 2018; Fig. 16c), whereas in our Model 3, the frontal thrust structures remain exposed and uncovered (Figs. 10, 11). Similar to our models, the syn-contractional sedimentation flanking the inverted syn-rift basins constrains the timing of inversion and thrust initiation, especially in the section through Las Aras, where apparent seismic reflectors onlap the buried
Triassic salt-detached thrust sheet (Fig. 16c). The thickness of the syn-contractional succession is sufficiently thick to support the inversion, uplift, and folding of the Las Aras basin in the hanging wall of the Las Aras thrust, and a syn-contractional footwall syncline develop. A section across the Las Aras basin shows some similarities to the section across minibasin 2 in our Model 3 with syn-contractional deformation (Figs. 10a and 16c) suggesting ideas for the interpretation of the basement sub-salt geometries.

One of the problems in the interpretation of the Las Aras section is the structural elevation difference between the bottom of the syn-contractional basin in the footwall of the inverted syn-rift basin and its equivalent above the syn-rift depocenter (Fig. 16c). The several km high structural relief has been interpreted as the result of basement underthrusting by the Axial Zone system (García-Senz, 2002). However, the timing of the main basement underthrusting occurred later (Upper Eocene-Oligocene) than the Upper Cretaceous (Muñoz et al., 2018), indicating that another mechanism is responsible for creating the
accommodation space for the kms thick succession of syn-contractional sediment in the footwall of Las Aras thrust. The accommodation for the thick syn-contractional depocenter could have partially resulted from the salt evacuation of a previously salt-inflated area south of the Las Aras basin, in the footwall of a basement-involved extensional fault. It is possible that the overturned limb of the footwall syncline could have developed above the short-cut of the extensional basement fault, as seen in Model 3 (Figs. 10a, c). The footwall shortcut and reactivation of the extensional fault could create additional accommodation
space for the Upper Cretaceous syn-contractional footwall syncline (Fig. 16c). To keep the cover system partially decoupled from the basement, Triassic salt would need to be preserved to facilitate the subsequent detachment and southern translation of the cover system.

In the South-Central Pyrenees, basement-involved shortening has been interpreted to occur after the development of the cover thrust system and is recorded by squeezing of pre-existing folds (Muñoz et al., 2018; Figs. 16b, c). However, model results
and the previous discussion above challenge this interpretation. We propose that basement-involved shortening, specifically the reactivation of pre-existing structures was coeval with the early development of the cover thrust system prior to its detachment and transportation to the south. In any case, in both the Pyrenees and our models, the basement thrust system is responsible for the regional uplift and geometry of the thrust wedge (Figs. 7a-c, 10a-c. 16), and the intervening salt layer keeps the cover thrust systems decoupled.

**Conclusions**

The results of the analogue modeling program presented in this paper provide geometric and kinematic insights of contractional systems that involve the inversion of extensional basins and their intervening accommodation zones with thick syn-rift salt. The extensional basement fault system controlled the location of overlying syn-rift depocenters and the evacuation pathways and inflation areas for the syn-rift salt layer. Synclinal minibasins developed above the hanging walls of the extensional
basement faults (Fig. 4) forcing salt to migrate away from the downbuilding basins. The salt flowed along fault planes and dipping basement panels updip into basement highs to form salt structures in the basement footwall blocks, in the hanging wall block along the margins of the syn-rift minibasins, and across the accommodation zone (Fig. 14a). Typical salt structures that develop are reactive to passive salt diapirs and salt anticlines which are generally perpendicular to the direction of extension;

diapiric salt walls and salt plugs can pierce the cover sequence where significant extension and syn-extensional erosion thin
the overburden (Fig. 4c). The salt structure across the accommodation zone tends to link the hanging wall salt anticline of one extensional fault to the footwall salt anticline of the adjacent fault via an oblique salt anticline (Figs. 4b, 6c). The synclinal minibasins extend beyond the confines of the basement half-grabens, obscuring the geometry of the underlying basement fault array. Primary welding of the minibasins occurs in the footwall and hanging wall, but not necessarily under the minibasins thickness maximum.

Upon shortening, the syn-rift salt kept the basement and cover thrust systems decoupled and exerted a fundamental control during inversion. The basement thrust system is characterized by low-angle thrusts and reactivation of the major extensional faults (Fig. 14b). An oblique ramp develops in the basement across the accommodation zone. The cover thrust system is characterized by source-fed thrusts that nucleate on preexisting salt structures, including primary welds that reactivate as thrust welds. The salt structures experience squeezing early and nucleate thrusts and folds. The addition of syn-contractional
sedimentation plays a key role in the propagation of deformation by delaying the transference of deformation into the foreland and possibly imposing the vergence of thrusts only towards the foreland. Syn-contractional sedimentation also increases the rate of salt extrusion from preexisting diapirs and pierced thrust tips.

The 3D implications of our results indicate the importance of the interaction of salt between individual basin systems in salt-bearing segmented rift systems. In extensional systems with sufficiently thick salt, the salt evacuation pathways from
downbuilding and extending minibasins can interact and modify the expected salt structure geometry, creating a salt ridge from the hanging wall of one extensional basin to the footwall of the adjacent basin via the accommodation zone (Fig. 14a). If the system then undergoes inversion and shortening, the distribution of inflated salt may control the location and style of thrusting in the cover sequence (Fig. 14b). This 3D understanding may provide insight and better templates for analyzing the trap potential for geological storage of $CO_2$, and production of geothermal energy in extensional to inverted salt-bearing rift
basins, such as the Lusitanian basin, offshore Portugal, and the Isàbena area of the South-Central Pyrenees.

**Acknowledgment**

This is a contribution of the Institut de Recerca Geomodels from the Universitat de Barcelona. We acknowledge the support of the SABREM (PID2020-117598GB-I00) research project funded by MCIN/ AEI /10.13039/501100011033 and the GEODIGIT (TED2021-130602BI00) research projects funded by MCIN/AEI/10.13039/501100011033 and for the European
Union "NextGenerationEU"/PRTR. We thank the financial support of OMV Exploration and Production through project FBG310629. Research by E. P. Wilson is supported by an AGUAR pre-doctoral grant (2020 FI SDUR) from the Generalitat de Catalunya. The Scientific colour maps *batlow*, *buda*, and *hawaii* from Crameri (2018) are used in this study to prevent visual distortion of data and exclusion of color vision deficiencies readers (Crameri et al., 2020).

**Supplementary material**

Time-lapse video of extension for Model 1.

Time-lapse video of shortening for Model 2 (no surface processes).

Time-lapse video of shortening for Model 3 (including syn-contractional sedimentation).

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
