# Peer review of "Inversion of accommodation zones in salt-bearing extensional systems: insights from analogue modeling"

_EGUsphere, 2022_

## Referee Comment (RC1)

Review of the manuscript

Inversion of transfer zones in salt-bearing extensional systems: insights from analogue modeling by

Elizabeth P. Wilson, Pablo Granado, Pablo Santolaria, Oriol Ferrer, Josep Anton Muñoz

Submitted to EGUsphere

The submitted manuscript aims to demonstrate the variation of structural styles in salt-bearing segmented rift systems that underwent subsequent shortening and syn-shortening sedimentation. The sand-polymer-based analog experiments involved three models, in which two salt-bearing half-graben basins segmented by an intervening transfer zone experienced rifting (Model 1), rifting and inversion of the rift system (Model 2), and rifting and inversion with syn-contractional sedimentation (Model 3). The obtained results of analog modeling are compared to the geological structures of the Northern Lusitanian Basin, offshore Portugal and Isàbena area, South-Central Pyrenees, Spain.

Experimental setup and procedure are well described. Preparing and conducting experiments likely required a substantial amount of work. The manuscript represents a thorough analysis of experimental results and provides a new insight into the salt decoupled extensional and inversion systems.

It would be helpful if the following comments and questions might be clarified:

Lines 110-115: why Mylar sheet remains not deformable under the simulation settings of extension and subsequent shortening? The basal friction at the top of the steel plate and Mylar sheet is different. Did it influence the model deformation?

Lines 120-130: Is there any difference in mechanical properties of the basement and upper sand packs? In sand material itself or in a way it was packed? See comment for lines 485-490.

Lines 125-130: what procedure was applied to ensure that the triangular polymer prisms would not be deformed during the model buildup and burial of the prisms by 9 cm-thick sand pack simulating the basement?

Lines 130-135, 140-145, 170-175: The applied combined velocity during extension phases was set equal to $2.78 \times 10^{-4}$ cm/s (phase 1) and $1.67 \times 10^{-4}$ cm/s (Phase 3) and to 1.67 x 10-4 cm/s during the shortening (Phase 5). Please explain what does "combined velocity" mean. Deformation of the polymer is sensitive to applied strain rate. How the velocity of extension and shortening was chosen? Was there any sensitivity study performed to determine the range of applicable strain rate?

What was sedimentation rate of sand layers during syn-contractional deformation?

Line 170: Please clarify here which wall was pushed - the one at the side of the Mylar sheet or of the metal plate. It seems to be the right wall with attached Mylar sheet that was used for shortening (Figs 8 and 11).

Line 180-185: what steps were undertaken to facilitate the models' slicing at intervals of 3 mm (!!) at the end of each experiment? Salt model polymer flows fast….

Line 200-205. Why MF1 attains more slip than MF2 in Model 1? The master listric faults MF2 and MF1 in Fig. 4a and 4c have different geometry, in particular degree of curvature and dip angle. Could these differences be related to the shape of triangle shape of salt model seeds in the basement? The seed SS2 after the extension generally preserves its triangle shape (Fig. 4a), while SS1 has migrated resulting in a weld between the basement sand layers and basal sheet (Fig. 4c).

Lines 240-250: "Half-graben 1 propagated along strike across the whole model width (Fig. 4 a-c). This is probably related to the presence of the underlying velocity discontinuity (V.D. in Fig. 2) that favors extension localization, lateral slip transfer along the strike of the polymer seed, and the formation of the largest depocenter of Model 1. Conversely, extension along MF2 produces a more diffuse structural pattern, with one largest depocenter right of the master fault".
Why MF2 and associated graben did not localize and propagate laterally to the same extent as MF1 and associated depocenter? Could it be that the behaviour of the V.D. between the rubber sheet and the Mylar sheet (MF2 system) was different from the V.D. between the steel plate and the rubber sheet due to the variation in the strength contrast between the basement materials: steel plate - rubber sheet - Mylar sheet? This suggestion is likely confirmed on lines 494-496.

Line 273: It is not mentioned here whether Model 2 was subjected to extension before the shortening. How steady are the results of extension obtained in Model 1 (Fig. 4)? Could it be expected that the results of Model 1 were repeated in Model 2 during the extension, and the shortening has started from the same or similar point as shown in Fig. 4?

Line 313. Replace SS2 by SS1 as Fig. 7c represents the section across the SS1 salt seed.

Line 310-315: If the results of extension in Model 2 repeated the results of Model 1, one could expect that the geometry of thrust fault systems in MF1 and MF2 during the shortening phase (Fig. 7a and 7c) would be controlled by the shape of listric faults MF1 and MF2 at the end of extension, which, in turn, might have been controlled by the behavior (degree of migration) of the model salt triangle seeds SS1 and SS2.

Models 2-3: Could the lower degree of shortening along MF1 be resulted from the greater distance between MF1 and movable backstop at the right, if compared to the shorter distance between the backstop and MF2? In both cases, most of the shortening is accommodated along MF2 that is located closer to the movable backstop, and the shortening structures progressed laterally across the model, accommodating most of the contractional deformation. As a result, shortening was more distributed in the section across MF1, and it contributed to reactivation of MF1 to lower degree.

Lines 485-490: "In our models, the syn-rift basin geometry and related sediment distribution are strongly controlled by the position of the underlying pre-salt basement faults, the amount and rate of slip attained by those faults, the original distribution and thickness of model salt, as well as the thickness and mechanical properties of the pre-kinematic sand pack".
However, in the model setup, nothing is said about the difference of mechanical properties of the basement and upper (pre-kinematic) sand packs (lines 120-130).

---

## Referee Comment (RC2)

[referee-annotated manuscript omitted]

---

## Author Response (AR1)

**Review of "Inversion of transfer zones in salt-bearing extensional systems: insights from analogue modeling"**

Authors: Elizabeth P. Wilson, Pablo Granado, Pablo Santolaria, Oriol Ferrer, and Josep Anton Muñoz

Dear editor and reviewers,

Thank you for taking the time and effort to review our manuscript. We sincerely appreciate your valuable comments, suggestions, and questions, which all help to improve the quality of the work. Our replies to each of Dr. Zwaan's comments and questions as well as the two reviewers are listed as bullet points beneath each original point.

We hope that our responses fully answer the questions and comments asked by the editor and the reviewers. We look forward to including our manuscript in his special issue on analogue modelling of basin inversion.

**Frank Zwaan (editorial committee)**

Some things to pay careful attention to:

**Editor:** Reviewer 2 points out that the research question is not very clear. I agree with that assessment: some topics are mentioned, but the reader needs to interpret a lot and it can be much more to the point.

> *Authors: The Introduction have been reorganized and text has been added to address these concerns and make the focus of the manuscript clearer.*

**Editor:** Linked to the previous comment: it is then not that clear why the parameters used in the set-ups are chosen as these are rather specific. It could for instance be useful to first (shortly) introduce the field examples in order to motivate the general study design, and/or weave these choices in the methods section as well (when mentioning a parameter in the set-up, you can explain where it comes from). This would be especially useful when introducing the different model phases.

> *Authors: We have introduced the main field example that has inspired the modelling program (South-Central Pyrenees) to justify the set-up design and explain the main differences with former studies such as Dooley and Hudec (2020) as well as the different model phases (two rift events and subsequent inversion).*

**Editor:** Scaling details seem to be missing, there is no explanation as to how the scaling values are obtained (formulas). Please add these somewhere in the manuscript.

> *Authors: Text has been added to the Table 2 caption.*

**Editor:** Specific details I noted:

**Editor, Fig. 2**: please indicate what is sand in the model. Now only the silicone is indicated, so it is not directly obvious what the cover is.

>*Authors: This has been updated.*

Velocities are given in cm/s, consider using cm/h for easy reading

>*Authors: The velocity has been converted to cm/h.*

**Editor:** Line 161-162: it is not 100% clear how this erosion works: what is the exact elevation benchmark for the scraping?

>*Authors: On line 164 it states that the regional sedimentation elevation was raised 1 mm for each new layer. The use of the scraper ensured that the elevation and distribution of each sand layer filled and covered the model at the correct elevation.*

**Editor:** Line 170: how exactly is the salt removed? i.e. is any salt that "sticks out" cut away? Perhaps add a sentence to specify

>*Authors: A sentence has been added to explain that any extruded salt that flowed onto the model surface was periodically removed by carefully cutting it away.*

**Editor:** Fig. 3: the normal faults in (c) and (d) are not that obvious. Consider making the lines a bit thicker to make them stand out a bit more.

>*Authors: This has been updated.*

**Editor:** Fig. 4: perhaps a little map-view inset could help to make the relations between the sections directly clear (without having to go to Fig. 5. Same for other figures.

>*Authors: Inset maps have been added to all 3 figures.*

**Editor:** Fig. 9 & 12: the thrusts are shown in red, but are poorly visible (to me, I have slight red-green colorblindness). Please consider another color.

>*Authors: I have attempted to choose a replacement color (similar to other colors used by Crameri, 2018, 2020) that will hopefully be more visible for readers that experience color blindness. Thank you for voicing this concern.*

**Editor:** Fig. 15: these plots seem incomplete? not each structure seen in the previous figures is provided? This figure also seems to present results, and as such would fit better in the results section. Perhaps it can be incorporated in Figs. 7 and 10?

*Authors:* *The plots have been added to their respective top view figures. Text has been added to the figure captions (Figs. 7 and 10), and text has been added to the results section for each model (paragraph 5, section 3.2; paragraph 6 section 3.3).*

**Editor:** Figure captions in general: please make sure that all abbreviations used in the figures are listed in their caption. This is not the case in various figure captions.

*Authors:* *Thank you. This has been corrected in all figures.*

**Editor:** In general, I would suggest avoiding abbreviations in the text, unless really necessary, as to promote readibilty (e.g. spell out minibasin instead of mb1).

*Authors:* *This has been updated in the text.*

**Review of "Inversion of transfer zones in salt-bearing extensional systems: insights from analogue modeling"**

Authors: Elizabeth P. Wilson, Pablo Granado, Pablo Santolaria, Oriol Ferrer, and Josep Anton Muñoz

**Elena Konstantinovskaya (Referee)**

**Reviewer:** The submitted manuscript aims to demonstrate the variation of structural styles in salt-bearing segmented rift systems that underwent subsequent shortening and syn-shortening sedimentation. The sand-polymer-based analog experiments involved three models, in which two salt-bearing half-graben basins segmented by an intervening transfer zone experienced rifting (Model 1), rifting and inversion of the rift system (Model 2), and rifting and inversion with syn-contractional sedimentation (Model 3). The obtained results of analog modeling are compared to the geological structures of the Northern Lusitanian Basin, offshore Portugal and Isàbena area, South-Central Pyrenees, Spain.

Experimental setup and procedure are well described. Preparing and conducting experiments likely required a substantial amount of work. The manuscript represents a thorough analysis of experimental results and provides a new insight into the salt decoupled extensional and inversion systems.

> *Authors: We appreciate these positive words.*

It would be helpful if the following comments and questions might be clarified:

**Reviewer:** Lines 110-115: why Mylar sheet remains not deformable under the simulation settings of extension and subsequent shortening?

*Authors: During extension, the mylar plastic is attached to the moving wall. After the extension and before the onset of shortening, the mylar is detached from the moving wall and fixed to the basal plate in order to detach the sand by moving the backstop wall over the mylar sheet.*

*Under the applied strain rate and forces, the strength of the mylar plastic sheet is great enough to keep it from being deformed. This is well documented in many previous published works. It is true that the mylar undergoes tension during the extension phase, but it does not undergo any significant changes in length. So, we can presume a small amount of elastic strain takes place but nothing significant to affect the modelling results.*

*Yet, to avoid misunderstandings, the "non-deformable (under modelling conditions)" statement has been removed and a much more detail description of the experimental procedure regarding the mylar sheet is provided in the second paragraph of section 2.2.*

**Reviewer:** Lines 110-115: The basal friction at the top of the steel plate and Mylar sheet is different. Did it influence the model deformation?

*Authors: It is not different since "We added a thin silicon layer (0.2 cm) underneath the sand pack completely covering both the steel plate, the rubber sheet and the mylar sheet" as currently stated in the main text and already explained in the figure caption of fig. 2 in the former version of the manuscript.*

**Reviewer:** Lines 120-130: Is there any difference in mechanical properties of the basement and upper sand packs? In sand material itself or in a way it was packed? See comment for lines 485-490.

*Authors: The mechanical properties and the sand material of the basement and upper sand packs are the same. The term 'basement' in our paper does not reflect any changes in rheological properties between the sand packages. Please see response to comment for lines 485-490.*

**Reviewer:** Lines 125-130: what procedure was applied to ensure that the triangular polymer prisms would not be deformed during the model buildup and burial of the prisms by 9 cm-thick sand pack simulating the basement?

*Authors: Procedure is now clarified in line 155.*

**Reviewer:** Lines 130-135, 140-145, 170-175: The applied combined velocity during extension phases was set equal to $2.78 \times 10^{-4}$ cm/s (phase 1) and $1.67 \times 10^{-4}$ cm/s (Phase 3) and to $1.67 \times 10^{-4}$ cm/s during the shortening (Phase 5). Please explain what does "combined velocity" mean. Deformation of the polymer is sensitive to applied strain rate. How the velocity of extension and shortening was chosen? Was there any sensitivity study performed to determine the range of applicable strain rate?

*Authors: "Combined velocity" has been defined in the fourth paragraph of section 2.2.*

*Yes, a series of tests were conducted to determine the possible velocities. No cover deformation was found at rates of extension below $2.22 \times 10^{-4}$ cm·s-1. We then tested velocities to determine which velocities worked best with our set up to create the types of structures that we are investigating. (This is now stated in the fourth paragraph of section 2.2)*

**Reviewer:** What was sedimentation rate of sand layers during syn-contractional deformation?

*Authors: The syn-contractional sedimentation rate was 2 mm of sand every 6 hours. This is now stated in the final paragraph of section 2.2.*

**Reviewer:** Line 170: Please clarify here which wall was pushed - the one at the side of the Mylar sheet or of the metal plate. It seems to be the right wall with attached Mylar sheet that was used for shortening (Figs 8 and 11).

*Authors: Yes, the wall used to push is the wall above the mylar sheet, as shown in Figure 2 and now clarified in the final paragraph of section 2.2.*

**Reviewer:** Line 180-185: what steps were undertaken to facilitate the models' slicing at intervals of 3 mm (!!) at the end of each experiment? Salt model polymer flows fast….

> *Authors: We have a workflow at the Geomodels lab using a specialized slicing machine that allows for rapid slicing and photography of the models. It is true that polymer flows quickly but the process was developed with this in mind to minimize the flow of polymer out of the model. We find there is no need to emphasize such specific part of the methodology here.*

**Reviewer:** Line 200-205. Why MF1 attains more slip than MF2 in Model 1? The master listric faults MF2 and MF1 in Fig. 4a and 4c have different geometry, in particular degree of curvature and dip angle. Could these differences be related to the shape of triangle shape of salt model seeds in the basement? The seed SS2 after the extension generally preserves its triangle shape (Fig. 4a), while SS1 has migrated resulting in a weld between the basement sand layers and basal sheet (Fig. 4c).

& Lines 240-250: "Half-graben 1 propagated along strike across the whole model width (Fig. 4 a-c). This is probably related to the presence of the underlying velocity discontinuity (V.D. in Fig. 2) that favors extension localization, lateral slip transfer along the strike of the polymer seed, and the formation of the largest depocenter of Model 1. Conversely, extension along MF2 produces a more diffuse structural pattern, with one largest depocenter right of the master fault".

Why MF2 and associated graben did not localize and propagate laterally to the same extent as MF1 and associated depocenter? Could it be that the behaviour of the V.D. between the rubber sheet and the Mylar sheet (MF2 system) was different from the V.D. between the steel plate and the rubber sheet due to the variation in the strength contrast between the basement materials: steel plate - rubber sheet - Mylar sheet? This suggestion is likely confirmed on lines 494-496.

> *Authors: In this section (section 3.1), as results, a purely geometrical description of the models is provided. Reviewer's accurate comments belong to the discussion, where we included, in former version of the manuscript, a discussion about this topic (see section 4.1, as stated in the last line of the comments). Still, in the current version and following reviewer's comments, few ideas regarding the different geometry of the master faults have been added.*

**Reviewer:** Line 273: It is not mentioned here whether Model 2 was subjected to extension before the shortening. How steady are the results of extension obtained in Model 1 (Fig. 4)? Could it be expected that the results of Model 1 were repeated in Model 2 during the extension, and the shortening has started from the same or similar point as shown in Fig. 4?

> *Authors: Yes, although this is stated in the model set-up and procedure we have included a "reminder" at the beginning of section 3.2.*

**Reviewer:** Line 313. Replace SS2 by SS1 as Fig. 7c represents the section across the SS1 salt seed.

*Authors: Yes, this has been corrected. Thank you.*

**Reviewer:** Line 310-315: If the results of extension in Model 2 repeated the results of Model 1, one could expect that the geometry of thrust fault systems in MF1 and MF2 during the shortening phase (Fig. 7a and 7c) would be controlled by the shape of listric faults MF1 and MF2 at the end of extension, which, in turn, might have been controlled by the behavior (degree of migration) of the model salt triangle seeds SS1 and SS2.

& Models 2-3: Could the lower degree of shortening along MF1 be resulted from the greater distance between MF1 and movable backstop at the right, if compared to the shorter distance between the backstop and MF2? In both cases, most of the shortening is accommodated along MF2 that is located closer to the movable backstop, and the shortening structures progressed laterally across the model, accommodating most of the contractional deformation. As a result, shortening was more distributed in the section across MF1, and it contributed to reactivation of MF1 to lower degree.

*Authors: We appreciate reviewer's comments. It is an interesting discussion that we have integrated in the second paragraph of section 4.2.*

**Reviewer:** Lines 485-490: "In our models, the syn-rift basin geometry and related sediment distribution are strongly controlled by the position of the underlying pre-salt basement faults, the amount and rate of slip attained by those faults, the original distribution and thickness of model salt, as well as the thickness and mechanical properties of the pre-kinematic sand pack".

However, in the model setup, nothing is said about the difference of mechanical properties of the basement and upper (pre-kinematic) sand packs (lines 120-130).

*Authors: Yes, this is a mistake. We meant the mechanical properties of the analogue model materials. This is an erratum and has been changed in the manuscript to state the 'analogue modelling materials (i.e., the sand and polymer)' in the first paragraph of section 4.1.*

**Review of "Inversion of transfer zones in salt-bearing extensional systems: insights from analogue modeling"**

Authors: Elizabeth P. Wilson, Pablo Granado, Pablo Santolaria, Oriol Ferrer, and Josep Anton Muñoz

**Anonymous Referee #2**

**General comments**

**Reviewer:** The manuscript presents results on a series of three analogue models of inversion basin tectonics including a viscous décollement layer (salt) and weak zones in the basement that simulate subsidenence at half-grabens in the basement that later also localize the thrust faults. The results of the study are clearly illustrated, and the manuscript is well written. The study is however very similar to another already published study of Dooley and Hudec (2020), who employed the same technique, and their map-view patterns and cross-sections and doubly-vergent thrusts are also similar, except the shape of the weak seeds in the basement layer that act as nuclei for developing normal and thrust faults.

The introduction mentions a wealth of literature, analog modeling studies on the inversion tectonics of salt basins, but I miss the definition of the clear objectives of the study. What are the unresolved scientific issues (e.g. fault and layer thickness patterns?) associated with inverted basins, especially those with segmented half-grabens? For example, how might the individual segments interact in terms of laterally migrating salt in the source layer? What are typical deformation patterns in the cover sequence in the segmented half-graben basins that might control on fault development and extrusion of salt in later stages (e.g. inversion)? This is difficult to imagine, becuase the Figure 1 only shows a single segment of the half-graben array. Are there any map examples that may have resulted in the inversion of the segmented half-graben basins? I am sure that some lessons about such systems can be derived also from the study of Dooley and Hudec (2020) - however there is no mention of this in the Introduction nor in the Discussion. How are your sets of experiments similar/different with respect to the latter study?

> *Authors: Additional text has been added in the second paragraph of the Introduction to summarize the modelling program of Dooley and Hudec (2020) and to differentiate our work from theirs. As stated in the text, there has been very little focus on the influence and role of extensional transfer zones in subsequent inversion of salt-bearing rift systems. Past research has focused on the inversion of the individual basins themselves, with little more than a passing remark on the contraction of the transfer zone. Even the Dooley and Hudec (2020) paper dedicates very little space to the description of inversion and contractional deformation through the transfer zones.*

If you consider the Aras and San Juan basins (discussion), as equivalent transects inside a larger inverted basin with segmented half-graben, what can we learn from the similarity of the structures in these two transects with the vertical profiles accross your models? Is this comparison even possible given the large translation of the cover sequence above the basement?

*Authors: It is only by using models in conjuncture with cross sectional restorations that will provide insights into the structural history of fold and thrust belts. The Las Aras and San Juan basins are interpreted to be incorporated into the same thrust sheet, and, therefore, to be translated in approximately the same position relative to one another. The thrust sheet has been translated away from its original position, relative to the basement, so the model results cannot be used as a present-day comparison to the Pyrenean example but can provide insights into the kinematics of inversion and controls on the resultant contractional geometries.*

**Response to comments in the annotated pdf -**

**Reviewer: Figure 1,** I wonder if this general figure is appropriate for the objective posed in the Introduction (line 75 - geometry and distribution of several depocentres). I suggest trying to draw a figure that would explain the possible mass transfer and cover interaction between two adjacent and laterally offset half-grabens.

**Authors:** *We only show a single half-graben in Figure 1 to highlight the fact that there has been little focus on the interaction of segmented basins in salt-bearing half-grabens; the example itself comes from a study that focused on the structural style of half-grabens related to relative amounts of decoupling of the sub- and supra-salt extensional systems (Richardson et al., 2005). Additional text has been added to the first paragraph of the Introduction to better explain the purpose and reason for the figure choice. While your comment is valid, the focus on mass transfer of syn-rift sediments is not a key focus of the paper as sediment accumulates via aggradation and not progradation.*
*This introductory figure was selected to familiarize the reader with a previous breakthrough in the role of a thick evaporite layer in a decoupled extensional system so that it can be compared to our results which focus on the interaction of decoupled and offset half grabens.*

**Reviewer, introduction:** For the Introduction paragraphs: What are unresolved questions regarding the natural salt bearing basins associated with their inversion stage and their potentially laterally segmented rift like original geometry? What are the examples from natural basins worldwide?

**Authors:** *The third Introduction paragraph has been modified to clarify better the unresolved questions regarding the influence of extensional transfer zones in the inversion of salt-bearing extensional systems. Additional text has also been added to the first two paragraphs to highlight the lack of previous works that focus on extensional transfer zones. As these zones are not typically the focus of studies, there are not many natural examples besides the studies sited in the first paragraph of the Introduction. We have also explained the main natural examples that have inspired our modelling program.*

**Reviewer, Lines 61-62:** ...in salt-bearing rift systems (Dooley and Hudec, 2020). Dooley and Hudec, 2020; specifically, carried out a very similar work to yours, their results should be outlined in the Introduction and reflected in the discussion, comparing with your models?

*Authors: Yes, while their models had a more regional scale by modelling a series of three segmented graben and does not explicitly discuss the influence of the transfer zones on contractional deformation, it is worth noting their results in the introduction. We have explained the main differences between our set-up and the one by Dooley and Hudec (2020) both in the Introduction and section 2.2*

**Reviewer, Lines 61-62:** I suggest to explain more specifically on how the placement of the seeds (weak points) changes the style of the basement deformation as it is critical for your approach, too.

*Authors: This is explained in greater detail in the following section (paragraph three of section 2.2). The shape of the seeds was chosen to constrain the basement fault geometry.*

**Reviewer, Line 91:** This viscosity number is wrong. It should be 1.6x10^4 ... (not -4). Correct in the scaling table and elsewhere in the text.

*Authors: Modified as suggested*

**Reviewer, Table 2:** wrong numbers for viscosity (see previous comment)

*Authors: Modified as suggested*

**Reviewer, Line 116:** is this correct word form? just "transfer" ?

*Authors: Transference is the correct word*

**Reviewer, Line 119:** domain with respect to

*Authors: Modified as suggested*

**Reviewer, Line 119:** in

*Authors: Modified as suggested*

**Reviewer, Figure 2:** the basal salt in the multilayer is missing in the schematic profile

*Authors: The basal salt is included, however, the thickness of the basal salt has been increased in Fig. 2 for better visibility*

**Reviewer, Figure 2, caption:** Explain in the caption and the text the purpose of the seeds.

*Authors: As previously commented, the purpose of the seeds is explained in lines 122-124*

**Reviewer, Figure 2, caption:** 7 cm in

*Authors: Modified as suggested*

**Reviewer, Line 159:** some word missing in sentence

*Authors: This refers to the regional elevation of sedimentation. It has been modified*

**Reviewer, Line 176:** p

*Authors: Modified as suggested*

**Reviewer, Line 159:** Or just "Profiles"? Can you explain the virtual inlines?

*Authors: Virtual inlines make a reference to cross lines and inlines as used when working with 3D seismic volumes. We use the term 'virtual' to indicate that the inlines result from software interpolation of the photographed cross sections taken of the model to create the voxel and resultant 3D segy file used for the three-dimensional interpretation of the results. To clarify the origin of the "virtual inlines", in the current paper we state "Virtual inlines extracted from voxels (Fig. 4 d to f) show...)"*

**Reviewer, Lines 517-518:** I do not understand, what dogma? Can you explain?

*Authors: The structural dogma defined by White et al. (1986) states that the cross-sectional geometry of the sedimentary fill in the hanging wall of a listric fault is directly related to the shape of the normal fault, assuming extensional deformation by simple shear. The text has been updated to better clarify this.*

**Reviewer, Line 707:** completely

*Authors: Modified as suggested*

---

## Author Response (AR2)

**Review of "Inversion of transfer zones in salt-bearing extensional systems: insights from analogue modeling"**

Authors: Elizabeth P. Wilson, Pablo Granado, Pablo Santolaria, Oriol Ferrer, and Josep Anton Muñoz

**Frank Zwaan (editorial committee)**

**Response to comments in the annotated pdf -**

**Editor, Line 55:** using "adressing how" would make the sentence flow a bit better

>    *Authors: The text has been updated.*

**Editor, Line 64:** "while" should be deleted it seems

>    *Authors: The text has been updated*

**Editor, Line 69:** by

>    *Authors: Yes, the text has been updated.*

**Editor, Line 73:** rifting stage/phase or so —> a word is missing it seems

>    *Authors: The text has been updated.*

**Editor, Line 74:** is the basement not by definition part of the extensional system? please rephrase

>    *Authors: Yes, we meant to reference the mantle exhumation. This has been modified in the text*

**Editor, Line 75:** only one rifted margin was involved (there was a refited margin on either side of the basin? I suggest using "rift basin" or so instead.

>    *Authors: Here we use rifted margin, and prefer to use the term, because we are referencing the regional geologic history, not just the individual basins from the Pyrenean natural case study.*

**Editor, Line 75:** the southern north sea and atlas were very small rift basins, so it should be stated that there are also variation in rift maturity.

>    *Authors: The text has been modified to include variation in rift maturity of these other examples.*

**Editor, Line 80:** analogue modelling

>    *Authors: Yes, this has been modified in the text.*

**Editor, Line 84:** I don't recall Brun & Beslier (1996) used seeds in their lithospheric-scale models? We did in Zwaan & Schreurs (2023, Tectonics), but perhaps a better citation is Le Calvez & Vendevelle 2002 in J. Virtual Explorer. Many other examples of studies using seeds exist.

> *Authors: You are correct. We meant Brun and Nalpas, 1996. This has been changed (as well as in the reference list). We have also added the Le Calvez & Vendeville, 2002 article as you suggested*

**Editor, Line 87:** consisting of

> *Authors: This has been updated in the text.*

**Editor, Line 88:** in which extension is followed by contrational deformation

> *Authors: This has been updated in the text*

**Editor, Line 90:** salt layer

> *Authors: This has been updated in the text.*

**Editor, Line 92:** does

> *Authors: Yes, this has been changed in the text.*

**Editor, Line 97:** there is a very nice review paper by Duffy et al. (2023) in the new journal of Tektonika, I suggest citing it here.

> *Authors: This reference has been added to the manuscript.*

**Editor, Line 98:** use "for storing hydrogen gas (H2)" —> hydrogen can also refer to the single atom (H)

> *Authors: This has been updated in the text.*

**Editor, Line 98:** natural gas (CH4)

> *Authors: This has been changed in the text.*

**Editor, Line 108:** this should be the Greek letter "mu"

> *Authors: This has been changed in the text.*

**Editor, Lines 119-120:** what is missing is a statement how time is scaled, an what the deformation velocties in the models would represent in nature. this is important when using materials with a strain-rate dependent rheology (such as the silicone polymer) —> it is important to provide the calculations and equations in the

text, so that the reader can check (and reuse) these. Part of this could potentially be moved to the appendix, but it is now lacking in the paper.

> *Authors: Table 2 has been updated to include the time and velocity scaling values and text has been added to the manuscript to explain the values.*

**Editor, Line 125:** delete?

> *Authors: The word has been removed from the text.*

**Editor, Lines 127-128:** the use of "wide" and "long" is a bit confusing, as "long" is normally used to indicate the longest side, which is not the case here. this seems to imply that the metal base is not large enough (35 cm, instead of the 50 cm model widht). I understand that "long" is meant to be in the x-direction (in map view), but it may be better to remove "long" and "wide" in this sentece, to avoid any confusion.

> *Authors: The words have been removed from the text.*

**Editor, Lines 129-134:** these sentences need a bit or rewording. It would be best to first state that extension is done by moving both backstops apart. Then, the mylar sheet is fixed to the base. Finally, one mobile wall moves inward (as I understand from the figure, the extension involves 2 moving walls, the inversion phase only 1 moving wall). The current way it is explained it a bit unclear.

Alternatively, as the whole procedure is explained a bit later on (so that there is some repetition): consider simply removing lines 131-133

> *Authors: We have opted to remove the repetitive text and leave the explanation in the following section.*

**Editor, Lines 137-138:** highlighted text

> *Authors: The highlighted text has been corrected and the unintentional paragraph spacing has been removed.*

**Editor, Line 138:** lateral = the long sides of the model?

> *Authors: No, we refer to the marginal area proximal to the moveable backstops. The text has been updated to clarify this.*

**Editor, Line 142:** Such a

> *Authors: The text has been modified.*

**Editor, Lines 145-146:** repreated statement, perhaps it is best to remove line 139-140.

*Authors: Agreed. The first sentence has been removed.*

**Editor, Line 147:** it is not 100% clear whether this highest part is the same in both models. It would be good to mention it somewhere. In Fig. 2a, it could be indicated with a dotted outline, to show it is the seed in the background (I hope that makes sense)

*Authors: The text has been modified and 'along their highest part' has been removed. We do not think that it is necessary to modify Figure 2 based on our changes to the text.*

**Editor, Line 151:** consists or consisted? Please check the text for consistency (either past or present tense, but not mixed)

*Authors: The text has been updated.*

**Editor, Line 153:** perhaps it is better to state that they are a means to localize listric faults (as is stated before)? Or that ther represented the \*impact\* of pre-exising weaknesses (the triangular shape does not seem to be something that we would expect in nature)?

*Authors: The text has been updated to better explain the purpose of the silicone seeds.*

**Editor, Line 160:** see comment on scaling: what does this velocity translate to in nature?

*Authors: The text has been updated.*

**Editor, Line 168:** it seems that this sentence should be part of the preceding paragraph? Is it separated on purpose? —> why is 9 cm chosen? (is there a link with the Pyrenees that inspired this work?

*Authors: It should be part of the previous paragraph and has been moved. 9 cm was chosen because it was the amount of extension required to create sufficient topography across the basement and create early half-grabens above the silicone seeds. There is no link to the amount of extension experienced by the Pyrenean rift system.*

**Editor, Line 173:** why slower? please explain shortly —> what is the link to the Pyrenees?

*Authors: The rate of extension is slower to prevent the model salt from behaving brittlely under strain, allowing it to flow and migrate in response to the developing supra-salt syn-rift basin system. It has no relation to any interpreted changes in the rate of extension in the Pyrenean example. The text has been updated to explain this change.*

**Editor, Figure 3:** the culumn in the figure shows a very nice definition of the various deposits. It would be good to use these definitions consistently in the text (especially in the dicussion). —> see some comments down the line.

*Authors: This has been updated.*

**Editor, Figure 3, Line 181:** it would be good to specify which section this is exactly, and to refer to the figure with the respective model results.

*Authors: These sections are used in the later figures. The caption has been updated to reference the corresponding section in figures 4, 7, and 10.*

**Editor, Line 187:** consider "and to prevent model salt extrusion" —> the sentence is a bit off

*Authors: The text has been modified.*

**Editor, Line 192:** delete

*Authors: Done.*

**Editor, Line 193:** this does not make sense: the layers are deposited at different times, one after the other.

*Authors: The layers were added consecutively without pauses for periods of downbuilding. The text has been updated to better explain this.*

**Editor, Line 196:** please provide a bit more motivation for including this tilt —> mentioning the South-Central Pyrenees

*Authors: Additional text has been added to explain its significance to the South Central Pyrenees (low angle unconformity).*

**Editor, Line 200:** delete

*Authors: The text has been updated.*

**Editor, Lines 201-202:** again, what is the link to the natural example? (what was the displacement and displacement rate in nature?)

*Authors: The text has been updated.*

**Editor, Line 206:** hoe much would this be in nature? (m/Myr?) would be good to specify

*Authors:  The text has been updated with calculated values added to Table 2.*

**Editor, Line 214:** I agree with reviewer 2 that it would be good to at least mention that an automated set-up is used for this (to make clear this is not simply done by hand, but by a more sophisticared method),

*Authors: The text has been modified.*

**Editor, Line 222:** 3D is used earlier on, may be best to choose one convention for the whole text

>*Authors: The text has been updated to only use 3D.*

**Editor, Lines 226-228:** It would be good to describe some of the key observations we can see in map view in the main text. E.g. I see in the sections (a) and (c) that two grabens did develop. I assume this is because they propagated parallel to each other, in a kind of rift pass strucutre (see e.g. Zwaan & Schreurs 2020 in Journal of Structural Geology for references on such rift interaction structures). This does not have to be long, but it would greatly help the reader understand the 3D structures a bit better —> I checked the video and it would just be nice to show the surface structures at the middle and end of the rifting phase in map view (as the upper row of Fig. 4, or Fig. 5).

>*Authors: A bit of text has been added to the first paragraph to briefly outline the 3D geometry of the system at the end of the model before moving on to the section interpretations.*

**Editor, Line 252:** perhaps add in the caption that this is an extension-only model so that the reader knows directly what is shown. Similar details can be added to the other figures showing such model results.

It would also be good to make clear in the caption that this is the end of the model run (similar detail should be added to the captions of the other section figures).

>*Authors: Additional text has been added to the captions in Figures 2 and 4-12 to remind the readers which model they are looking at.*

**Editor, Line 258:** it could be useful to make this the start of a sub-section within section 3.1 (including a header), as it is mostly about the evolution of the model, whereas the previous text is mostly about the final model strucutre. This could be helpful to the reader.

>*Authors: We would prefer not to add a sub-section here. The last three paragraphs of the section describe the final basement fault organization as well as the distribution of the model and salt and syn-rift succession and wouldn't fit into a sub-section about the model evolution. Additionally, as this was not suggested for models 2 and 3, it would be strange if all the model results did not follow the same format.*

**Editor, Line 279:** see previous comment on showing the evolutions of the initial structures in map view.

>*Authors: We do not understand what you mean. There is a small inset top view image to show the location of the sections and the following 2 figures and text describe the evolution of the model and final structure and true stratigraphic thickness maps of the final configuration of the various units.*

**Editor, Line 283:** due to the VD I assume?

*Authors: Correct. Text has been added to reflect this.*

**Editor, Line 284:** The term "transfer zone" is used in this paper. I would suggest using "accomodation zone" instead, as there is no "hard linkage" between both grabens (to me, a transfer zone is a linkn between two rift basins in the shape of through-going faults, which is not the case here). See for instance Zwaan et al. (2016) in Tectonophysics, and the references therein regarding the relevant terminology. A potential neutral option could be "rift linkage structures"

*Authors: The text has been updated and all "transfer zone" terms have been replaced by "accommodation zone", to the editor's preference.*

**Editor, Line 303:** what is indicated are not grabens, but normal faults?

*Authors: The figure has been modified so that the arrows point to the space between the lines so that they fall within the grabens.*

**Editor, Line 306:** for clarity: please specify that this is at the end of the model run/extension phase

Same for other map view imagery, where relevant

*Authors: This has been updated on all figures.*

**Editor, Line 439:** "despite being" might work better

*Authors: The authors would prefer to stick with the current verbiage. The use of the word 'being' makes it sound as though the post-salt sequence is welded to the basement, but we have no way of knowing where the system was or was not welded prior to shortening.*

**Editor, Line 529:** I believe it can (should?) be written together: "backthrust" —> consider doing so for the whole manuscript.

*Authors: The text has been updated following the editor's preference.*

**Editor, Line 548-550:** I am not sure about this argument. As far as I understand, the mylar sheet is rigid, so I would the impact along the "mylar VD" not expect to be that different from that along the "metal VD". Instead, I suspect it could have to do with the shape of the seeds —> the tip, i.e. that weakest spot in the model, is located towards the "metal VD" in both cases. So there is likely a difference in stresses on both sides, and deformation as such could be more foucces on the "metal VD" side.

*Authors: Yes, your point makes sense as the three basal materials are rigid. However, the metal plate has a greater thickness than the comparatively paper-thin (only use qualitatively as a measurement comparison) mylar sheet. And this greater vertical offset between the metal plate*

*and rubber sheet, versus that of the mylar sheet to rubber sheet, is what helps make it a more effective VD. Additional text has been added to the discussion to address your point as well.*

**Editor, Line 553:** akin to a rift pass structure —> see previous comment on this topic.

*Authors: As stated previously, transfer zone has been replaced by accommodation zone throughout the text.*

**Editor, Line 573:** "the dogma that states that the" —> given the wealth of knowledge on decoupling due to salt, is it really fair to present this dogma as a on-going issue? Perhaps it can be rephrased a bit.

*Authors: The text has been modified and the term has been removed.*

**Editor, Line 593:** concerns

*Authors: This has been replaced*

**Editor, Line 605:** 14a?

*Authors: Correct. Thank you for catching this.*

**Editor, Lines 630-631:** I suspect this effect may have been observed in previous models as well —> it would be nice to link it to a previous paper (I believe Jara et al. 2015 or 2018 could be of interest).

*Authors: References have been added.*

**Editor, Line 638:** this paragraph is rather long (a "wall of text") so that it becomes a bit hard to follwo. To improve readability, it would be better to split it up in 2 or three paragraphs.

*Authors: Agreed. It has been split into 3 paragraphs.*

**Editor, Lines 661-662:** here a citation is needed, and a specification of what is expected instead.

*Authors: The text has been updated*

**Editor, Line 670:** in

*Authors: This has been corrected.*

**Editor, Lines 671-672:** Is there no previous work that has looked a bit into this effect within the current context? I remember that Leonardo Pichel may have done some experiments with tilted basement and sedimentation loading (in salt-bearing systems?). If so, it should be cited.

In general, the stabilizing effects (more localized fautling) of a thicker salt overburden is in line with previous (modelling) works, and a short comment + some citations on this would be good to include here.

*Authors: We have added citations for previous works that discuss this topic.*

**Editor, Lines 676-677:** consider "of a salt-bearing system of which the overburden has been detached"

*Authors: The text has been modified.*

**Editor, Line 676-677:** not only rifted margins, but also for instance the Jura Mountains —> see for instance the work by Marc Schori (2019) in tectonophysics and in general the work done by the team of Jon Mosar in Fribourg (Switzerland). The transport of the overburden over many kiliometers is of crucial importance for the interpretation of the system.

*Authors: Additional text has been added to the end of the paragraph to make this point.*

**Editor, Line 682:** as commented by reviewer 2, this should be "transfer" instead of "transference", please check throughout the text. —> transference seems to be a term that is used to describe the transfer of nowledge or behaviour to other people

*Authors: Transfer is typically the verb form of the word (while also used as the noun form) while transference is a noun form of the word as is ok. However, since only transfer is used in the rest of the text, the term has been exchanged here as well.*

**Editor, Line 716:** these can be deleted I'd say?

*Authors: The text has been updated.*

**Editor, Line 729:** it seems that in the new interpretation, the top basement has a different geometry from the Alves et al. (2002) interpretation. Why is this? Is the basement not clearly defined on the seismic line to allow for this difference?

*Authors: Yes. This has been explained in the text. But the text hase been modified to highlight this.*

**Editor, Line 730:** note the typo in the figure header (et a., 2002)

*Authors: This has been corrected.*

**Editor, Line 743:** basins?

*Authors: No. But the text has been modified to 'syn-rift basin as a salt-detached ramp syncline basin' to prevent further confusion.*

**Editor, Line 759:** delete?

    *Authors: It is not necessary to delete the 'the'*

**Editor, Line 775:** please indicate the location of the basin on the map in Fig. 16 (and in the sections as well) —> is it all the post-rift? (yellow and pinkish colours?)

    *Authors: Thank you for catching this – the Cotiella basin is the large expanse of dark green on the western edge of the map. A label has been added to the map and clarified in the section.*

**Editor, Lines 786-789:** that is, you propose that in the original situation, the basins were situated below the Las Aras and San Juan Basin, but that now all has been collectively transported some 10's of kilometers to the south? I think this needs to be specified a bit clearer as the current text seems to imply the original basins are below the current position of the LA and SJ basins.

    *Authors: We mean the supra-salt cover system comprised of the pre-kinematic-post-rift successions and syn-rift basins have been detached and translated to the south, separately from the basement rift system. The basement rift system is incorporated into the Axial Zone. The text has been updated to clarify this.*

**Editor, Line 791:** that is, the transfer zone (or accomodation zone) between the thrust system? The original transfer zone should be situated far to the north?

    *Authors: No the pre-existing extensional transfer zone is positioned to the NW of the Las Aras basin between it and the San Juan basin. The text has been updated to clarify this.*

**Editor, Line 793:** where is this basin? please indicate

    *Authors: The syn-contractional succession has been better defined in Figure 16 and the text has also been updated.*

**Editor, Line 793:** can this be quantified? please add some citation as well

    *Authors: Instead of providing sedimentation rates, we have better described the differences in final syn-contractional sediment distribution to illustrate our point. The thrusts in the South-Central Pyrenees have been completely buried by the syn-contractional sedimentation, whereas in our Model 3, the sedimentation rate is not high enough to bury the thrust structures.*

**Editor, Line 796:** it is not that clear what this means? sufficiently strong? thick? in what way does it support? do you mean "allow"?

    *Authors: We mean sufficiently thick and this has been added to the text.*

**Editor, Line 801:** which thrust is that? please annotate in the figure. Otherwise, the text is rather difficult to follow —> I assume synorogenic is syn-contractional? I suggesting using a single term here.

> *Authors: Yes. Thank you for catching this. It should read syn-contractional and has been updated throughout the text. A label for the Las Aras thrust has been added to Figure 15.*

**Editor, Line 804:** consider "have partially be the result of salt evacuation from a previously salt-inflated area"

NB: some words in the text should perhaps be hyphenated when they are used as adverbs (e.g. salt-inflated). I suggest checking the text for these instances

> *Authors: The text has been modified.*

**Editor, Line 807-809:** I had some trouble following this. further to the north and south of what? —> of the footwall syncline?

This stems a bit from the fact that it is not that clear which structure in the figure is referred to (see comment on annotation).

In order to allow for easy comparison between Fig. 10a and Fig 16c, it may even be a good idea to include the model section in Fig. 16, directly below the seismic section interpretation

> *Authors: Yes, this is rather difficult to follow. The text has been modified more clearly state our point and compare the section to Model 3 results.*

**Editor, Line 810:** this sentence is unclear, is this an interpretation made by others? if so, provide citations

> *Authors: The text has been updated to include a citation.*

**Editor, Line 810:** remove "unlike out models"? this point is made in the next sentence, so this seems a bit repetitive

> *Authors: The text has been updated.*

**Editor, Line 812:** how does it challenge this interpretation? please specify to hammer the message home: perhaps add somehting like: "instead, we propose that basement-involved shortening was coeval with the developement of the cover thrust system"

NB: if this is a major point to be made, it should be mentioned in the introduction as well.

> *Authors: A sentence has been added to better state the message.*

**Editor, Line 817:** please mention the transfer zone as well, as that is in the title, and thus a key part of the paper.

*Authors: The text has been updated.*

**Editor, Line 843:** why not mention the natural examples discussed in the text to make this point a bit more concrete?

*Authors: The text has been updated.*

---

## Author Response (AR3)

**Review of "Inversion of transfer zones in salt-bearing extensional systems: insights from analogue modeling"**

Authors: Elizabeth P. Wilson, Pablo Granado, Pablo Santolaria, Oriol Ferrer, and Josep Anton Muñoz

We would like to thank the editor for his work and thoughtful comments, suggestions, and corrections to our manuscript that help make our work better.

**Frank Zwaan (editorial committee) -**

**Public justification (visible to the public if the article is accepted and published):**

Dear authors,

Many thanks for your swift and detailed reply, and for the resubmission of the updated manuscript.

Congratulations on this nice modelling study. I only have a couple of minor suggestions for small corrections to the current version of the manuscript that I would like to ask you to quickly check, before I can forward your work to the executive editor for publication (see details in the uploaded PDF file).

Looking forward to receiving the final version of your manuscript,

Best regards,

Frank Zwaan

**Response to comments in the annotated pdf provided by the editor -**

**Editor, Line 1:** you could perhaps consider using for instance "offset half-grabens" instead of "accomodation zones" here. the grabens seem to be the focus of the paper, rather than the accomodation zones?

(see also highlighed terms in the abstract, "accomodation zone" only comes in rather later in the abstract)

> *Authors: We would prefer to keep the title with the focus on accommodation zones, as one of the key points of our work is that these structures have not been typically been the focus of recent papers, since their definition in the 1980s-1990s. The term accommodation zone has been added to the first sentence so that it is mentioned earlier in the abstract.*

**Editor, Line 8:** consider using "offset" instead of segmented, which is a bit vague (to me) —> using "offset" clearly says the grabens are separated and not oriented in a continuous fashion

*Authors: We would prefer to continue using the term, as it is the term used by Dooley and Hudec (2020) amoung others to describe en-echelon rift systems segmented by transfer zones*

**Editor, Line 14:** perhaps use "accomodation zone between both half-grabens" for clarity

*Authors: The text has been modified.*

**Editor, Line 76:** again, the Atlas and North Sea are not margins, but the text seems to imply they are —> I suggest using "extensional systems" or something similar (perhaps "rifted margins and rift systems"? or "rifts and rifted margins"?)

*Authors: The text has been updated.*

**Editor, Line 78:** see previous comment on the use of margins

*Authors: The text has been updated.*

**Editor, Line 84:** I checked Brun & Nalpas (1996), and I believe this reference should be removed here. The reason: a seed is normally used to localize deformation. In Brun & Nalpas (1996) the silicone patches work in the opposite way: they distribute the otherwise strong localization of the basal velocity discontinuity. Also, the paper does not involve any offset rift system as in this paper (and the other papers cited in this sentence). A better citation would be Schmid et al. (2023) in Solid Earth.

*Authors: The reference has been removed and the text has been updated.*

**Editor, Lines 567-568:** it would be good to specify these thicknesses somewhere (3 mm vs <1 mm or so?, it can just be put in parantheses —> but it would also be good to shortly specify this in the methods, for completeness

*Authors: The text has been updated and these values have also been added to the methods section.*

**Editor, Lines 705-706:** double use of "such as" —> perhaps just delete "such as salt basins and passive margins"? (also since the Jura and Alps are mountain ranges, even though they may contain/be built on previous basins).

*Authors: The text has been updated.*

**Editor, Line 794:** it seems that the San Juan basin is missing in this first list of basins?

*Authors: The text has been updated.*

**Editor, Line 814:** just spell out "west" ?

*Authors: This has been updated in the text.*

**Editor, Line 817:** kinematic

*Authors: Thank you. This has been corrected.*

**Editor, Line 818:** perhaps use "salt" as is done elsewhere?

*Authors: This has been updated.*